# Decoupling Semantic Similarity from Spatial Alignment for Neural Networks

**Tassilo Wald**[*,1,2,3] , **Constantin Ulrich**[1,4,7] , **Gregor Köhler** [1,3] ,
**David Zimmerer** [1,2] , **Stefan Denner** [1,3] , **Michael Baumgartner** [1,3] ,
**Fabian Isensee** [1,2] , **Priyank Jaini**[†,5] , **Klaus H. Maier-Hein**[†,1,2,3,4,6]

[1] Division of Medical Image Computing,
German Cancer Research Center (DKFZ), Heidelberg, Germany
[2] Helmholtz Imaging, DKFZ, Heidelberg, Germany
[3] Faculty of Mathematics and Computer Science,
University of Heidelberg, Germany
[4] Medical Faculty Heidelberg, University of Heidelberg, Germany
[5] Google Deepmind
[6] Pattern Analysis and Learning Group, Department of Radiation Oncology
[7] National Center for Tumor Diseases (NCT) Heidelberg, Germany

## Abstract

What representation do deep neural networks learn? How similar are images to each other for neural networks? Despite the overwhelming success of deep learning methods key questions about their internal workings still remain largely unanswered, due to their internal high dimensionality and complexity. To address this, one approach is to measure the similarity of activation responses to various inputs. Representational Similarity Matrices (RSMs) distill this similarity into scalar values for each input pair. These matrices encapsulate the entire similarity structure of a system, indicating which input leads to similar responses. While the similarity between images is ambiguous, we argue that the spatial location of semantic objects does neither influence human perception nor deep learning classifiers. Thus this should be reflected in the definition of similarity between image responses for computer vision systems. Revisiting the established similarity calculations for RSMs we expose their sensitivity to spatial alignment. In this paper, we propose to solve this through *semantic RSMs*, which are invariant to spatial permutation. We measure semantic similarity between input responses by formulating it as a set-matching problem. Further, we quantify the superiority of *semantic* RSMs over *spatio-semantic* RSMs through image retrieval and by comparing the similarity between representations to the similarity between predicted class probabilities.

## 1 Introduction

Deep neural networks are trained to extract powerful feature representations for a wide range of downstream tasks. Despite this, their inner workings are highly-complex, making understanding *how* networks solve tasks and *what* they learn challenging. To obtain a better understanding of these fundamental questions, researchers in the fields of neuroscience,

---

[*]Corresponding author: tassilo.wald@dkfz-heidelberg.de
[†]Shared last authorship.

38th Conference on Neural Information Processing Systems (NeurIPS 2024).

cognitive science, and machine learning independently developed various methods to interpret and relate representations [29].

In the realm of machine learning, a lot of prior-methods have been proposed to meaningfully measure the similarity between intermediate representations of ANNs [15, 31, 25, 19, 12]. Klabunde et al. [10] provides a comprehensive summary and categorizes them into measures based on a) Canonical Correlation Analysis (CCA) [25, 19] b) Alignment measures c) Representational Similarity Matrices (RSMs) d) Nearest Neighbors e) Topologies and f) Descriptive Statistics.

Of all these methods, Representational Similarity Matrix (RSM) [13] based measures have enjoyed the most attention over the last years. RSMs consist of sample-to-sample comparison, measuring the similarity between the (intermediate) responses of the same network to two different samples. Based on many such comparisons, the RSM represents the similarity structure of what a model considers similar. This representation of a system's behavior reduces the highly-dimensional, complex internal structure, that the model of interest may possess, to a $N \times N$ Matrix for $N$ samples. It enables the comparison of the similarity structure of any system, as long as one can input the same samples and measure the similarity between the responses.

In the machine learning domain, this concept was introduced by Kornblith et al. [12] in conjunction with Centered Kernel Alignment (CKA) to compare the similarity between RSMs of different layers within a model or across models. CKA superseded previously popular Canonical Correlation Analysis (CCA) metrics [25, 19], since they need a vast amount of samples to measure similarity. Consequently, CKA was used in various applications, to measure the similarity between Transformers and CNNs [26] or wide and deep networks [21] and to understand catastrophic forgetting [27] or transfer learning [20] to name a few.

In this paper, we revisit the key component of the most used similarity measure in the field: Representational Similarity Matrices and how they are constructed in the vision domain. The key contributions of our work are summarized as follows:

- We highlight that current RSMs are constructed in a way that couples localization and semantic information, which constraints one only to measure similarity if, spatial and semantic information aligns between two samples.

- To address this issue we propose *semantic RSMs*, which are invariant to spatial permutation and exclusively measure semantic similarity, by formulating it as a set-matching problem.

- We show that the inter-sample similarity of semantic RSMs leads to improved retrieval performance and better reflects the similarity between representations of classifiers and their predictive behavior.

- Moreover, due to the computational complexity of the proposed algorithm, we introduce approximations that significantly reduce computation time.

The Code is available here.

## 2 Representational Similarity

**Formalization** To establish the concept of representational similarity in the context of computer vision, we provide a brief formalization of the problem. Let $x \in \{x_0, \ldots, x_N\}$ denote the input samples for which we collect input responses $Z_1 \in \{z_{1,0}, \ldots, z_{1,N}\}$ and $Z_2$, which we call representations. The representations can take different shapes, with $Z_{\text{CNN}} \in \mathbb{R}^{N \times C \times W \times H}$ denoting the responses of a CNN with $C$ channels and a spatial extent of $W, H$ or $Z_{\text{ViT}} \in \mathbb{R}^{N \times D \times T}$ denoting the responses of a ViT with depth $D$ and tokens $T$. For the purpose of simplification and without loss of generality, we unify the spatial dimensions $W, H$ for CNNs and $T$ for ViTs into a joint spatial dimension $S$, resulting in $Z \in \mathbb{R}^{N \times C \times S}$. Given representations $Z$, we can construct RSMs $K, L \in \mathbb{R}^{N \times N}$, with values $K_{ij} = k(z_{1,i}, z_{1,j})$ and $L_{ij} = l(z_{2,i}, z_{2,j})$ measuring how similar the representation $z_i$ is to $z_j$ given the kernels $k$ or $l$. The kernels define the measure of similarity between the

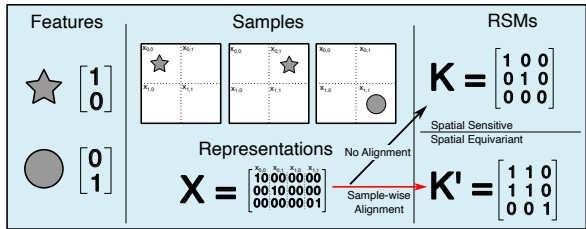

Figure 1: Current spatio-semantic RSMs couple semantic similarity with spatial alignment. Our proposal focuses solely on measuring semantic similarity. We achieve this by determining the optimal permutation between two representations and introducing sample-wise permutation invariance.

representation vectors and hence play an important role. We introduce an exemplary kernel in Section 3 and all kernels used in this paper with some properties in Appendix A.

With the two RSMs $K$ and $L$ at hand, it is possible to compare the similarity structure between the two models. As introduced in Kornblith et al. [12] one can use the Hilbert-Schmidt independence criterion (HSIC) [5, 28] to calculate the level of independence through Centered Kernel Alignment, providing a measure of similarity of the two representations $Z_1$ and $Z_2$ with $\mathcal{H}$ denoting the centering matrix.

$$\text{HSIC}(K, L) = \frac{1}{(n-1)^2}\text{tr}\left(K\mathcal{H}L\mathcal{H}\right) \tag{1}$$

$$\text{CKA}(K, L) = \frac{\text{HSIC}(K, L)}{\sqrt{\text{HSIC}(K, K)\text{HSIC}(L, L)}} \tag{2}$$

Alternatively, a variety of different measures based on RSMs are possible for which we refer to Section 3.3 of Klabunde et al. [10]. As highlighted above, the similarity calculation based on RSMs is a two-step process with the first being the calculation of the RSMs and the latter being the comparison of the RSMs. In this paper, we focus on the first step, by quantifying the importance of disentangling semantic similarity from spatial alignment. While not the focus of this paper, we provide qualitative examples of the downstream effect on CKA measures in Appendix G.

## 3 The Semantic Representational Similarity Matrix

Representational Similarity Matrices (RSMs) are designed to reflect the system behavior of interest. The RSM $K$, originally introduced by Kriegeskorte et al. [13], represents the similarity structure of a system given a set of inputs $x \in \{x_0, \ldots x_N\}$. Each value $K_{ij}$ in $K$ quantifies how similar the responses of two inputs $z_i$ and $z_j$ are to each other. The definition of what symmetries between representations similarity measures should be invariant to is a central point of debate. Previous work proposed permutation invariance [15], invariance to orthogonal transformations [12], or invariances to invertible linear transformations [25, 19]. While arguments for any of these invariances are valid, we believe that an important aspect has been neglected in the calculation of RSMs: The spatial alignment between the representations!

**The dependency on spatial alignment**  Revisiting the structure of representations of a CNN, channels $C$ correspond to semantic concepts while the spatial position corresponds to where the semantic concept is localized in the input image [34]. Consequently, one can reformulate the representation $z_i$ of a sample $x_i$ to be fully defined by a set of **semantic concept vectors v**, one for each spatial location $S$: $z_i = \{\mathbf{v}_0, \ldots, \mathbf{v}_S\}$ with $\mathbf{v} \in \mathbb{R}^C$. In the case of linear CKA [12], the RSMs are then calculated, between semantic concept vectors at the same spatial location. For instance, when employing the linear kernel $K_{ij}$ can be expressed as:

$$K_{ij} = \sum_s \langle \mathbf{v}_{z_1,s}, \mathbf{v}_{z_2,s} \rangle \tag{3}$$

This formulation emphasizes the coupling of semantic similarity and localization during similarity calculation, which we term as *spatio-semantic RSMs*. This coupling can lead to issues, e.g. when comparing an image to a translated version of itself. Due to the quasi translation-equivariant nature of CNNs[3] semantic vectors are translated similarly, changing the alignment of pairs of $\mathbf{v}$, leading to a low perceived similarity despite highly similar semantic vectors. This issue is visualized in a small toy example in Fig. 1

### 3.1 Decoupling Localization and Semantic Content

As shown above, current RSMs compare different input samples without accounting for the lack of spatial alignment. Previous work of Williams et al. [32] recognized this and introduced translation invariance to RSMs by finding the optimal translation $a, b$ of the representations $z'_j = \{\mathbf{v}_{0+a,0+b}, \ldots, \mathbf{v}_{w+a,h+b}\}$ to maximize similarity $K_{ij} \quad \max_{a,b} = \langle z_i, z'_j \rangle$ through circular shifts.

While this is an improvement to no spatial alignment and emulates a CNN's inherent translation equivariance, we argue that the measure of representational similarity should not be constrained to what the underlying model is invariant to, but the similarity measure should be invariant to the possible spatial configurations of semantic features in the input image.

To motivate this, we propose a thought experiment:
Imagine we have trained a classifier with an augmentation pipeline including rotations. Given an image and a rotated version of the image, we extract representations $z$ at layer $i$ once for the normal $z_i$ and once for the rotated image $z_{i,rot}$. Due to the initial rotation, these representations may differ in earlier layers $i$, due to the network extracting different edges and corners. However, if the network successfully learned to become invariant to the augmentation, it may have learned to map it to the same semantic vector $\mathbf{v}$ at a later layer but at a different spatial location. For such cases, we argue that the similarity between the two representations should be high. Should the model be sensitive to the rotation, no semantically similar representations may be expressed at a later layer, which should lead to a low similarity.

This reasoning can be extended to all kinds of shifts, be they artificial augmentations like shearing or mirroring or natural variations of the input manifold. **Subsequently, we argue that the similarity measure should be invariant to as many spatial shifts as possible. This alone allows one to measure the similarity of representations a model is invariant to, be these learned or designed invariances.** Such variable shifts cannot be captured with simple translation operations.

#### 3.1.1 Introducing permutation invariance

To impose as minimal constraints on spatial structure as possible, we propose to make $K_{ij}$ invariant to all spatial permutations of the semantic concept vectors $\mathbf{v}$. Formalizing this we demand that the similarity $K_{ij} = k(z_i, z_j) = k(z_i, \mathbf{P}_{ij} z_j)$ with $\mathbf{P}_{ij} \in \mathbb{R}^{S \times S}$ being a unique permutation matrix for the pair of $z_i$ and $z_j$. To accomplish this, we propose to find the optimal permutation matrix $\mathcal{P}_{ij}$ that maximizes the similarity $K_{ij}$.

$$\mathcal{P}_{ij} = \text{argmax}_P \quad k(z_i, \mathbf{P}_{ij} z_j) \tag{4}$$

To find the optimal permutation matrix $\mathbf{P}_{ij}$, we decide to use the linear kernel $\langle \cdot, \cdot \rangle$ to maximize both the magnitude of activation and the direction of vectors, as both magnitude of activation and direction of the vectors matter[12]. This allows us to calculate an affinity matrix $\mathbf{A}_{ij} \in \mathbb{R}^{S \times S}$ measuring the similarity between all concept vectors:

$$\mathbf{A}_{ij} = [\mathbf{v}_{i,0}, \ldots \mathbf{v}_{i,S}]^{\mathsf{T}} [\mathbf{v}_{j,0}, \ldots \mathbf{v}_{j,S}]. \tag{5}$$

With this affinity matrix, bipartite set-matching algorithms, such as Hungarian matching, can be employed to find the optimal permutation matrix $\mathcal{P}_{ij}$ that maximizes the inner product between $z_i$ and $z'_j$. Finding all $\mathcal{P}_{ij}$ for all pairs $i, j$ and applying the chosen kernel $k$ yields the *semantic RSM*. This *semantic RSM* is invariant to any arbitrary, unique spatial

---

[3]The patch embedding can similarly translate semantics to a different position in the sequence.

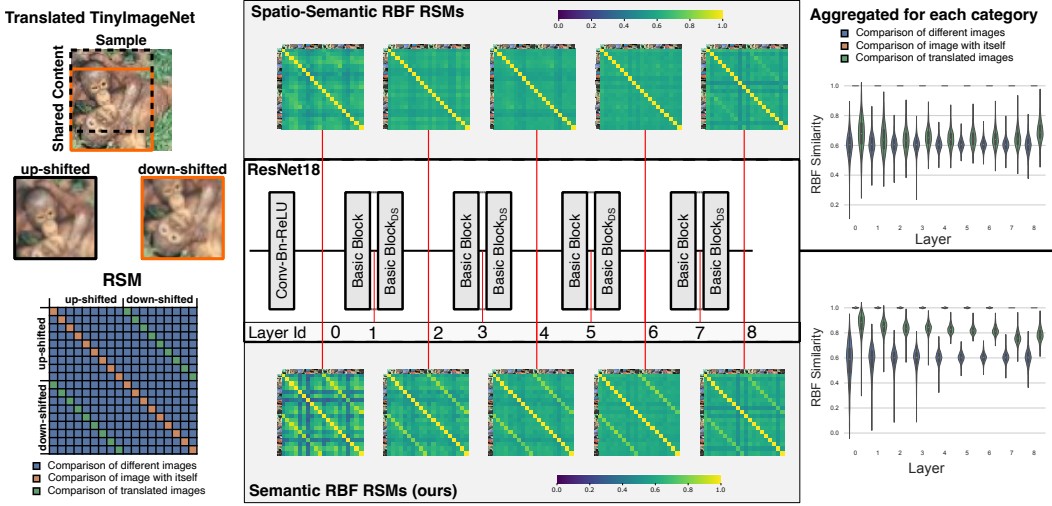

Figure 2: **Semantic RSMs capture similarity independent of spatial localization, in contrast to current spatio-semantic RSMs.** We utilize Tiny-ImageNet to generate partially overlapping crops of the same sample (left) and calculate RSMs for a trained ResNet18 model. The plot displays the original spatio-semantic RSMs (middle top) and our proposed semantic RSMs (middle bottom) across various layers for a single batch. Additionally, the distribution of similarity values over multiple batches is shown (right). The results indicate that spatio-semantic RSMs struggle to detect largely identical but translated images, while semantic RSMs exhibit an enhanced off-diagonal in the RSMs and a significant gap between distributions. This demonstrates the capability of our method to detect the same semantics even when translated.

permutation $\mathbf{P}_{ij}$ for each pair of representations, and, depending on the choice of kernel $k$, invariant to orthogonal transformations $U \in \mathbb{R}^{C \times C}$ along the channel dimension. These *semantic RSMs* can be used as a drop-in replacement for any other RSM, e.g. for applications such as calculating $\text{CKA}(K, L)$ to measure the similarity between systems.

**Computational Complexity**    Finding the optimal permutation matrix $\mathcal{P}_{ij}$ is NP-hard and needs to be repeated for each pair of representations $z_i, z_j$. With $N$ samples, this results in $\frac{N \cdot (N+1)}{2}$ unique permutations that need to be computed for a *semantic RSM*. The overall complexity of bipartite matching algorithms grows with the spatial dimensions cubed, resulting in $\mathcal{O}(N^2) \times \mathcal{O}(S^3)$. The outer $\mathcal{O}(N^2)$ complexity can be parallelized, or reduced by decreasing the batch size. However, the inner permutation can become time-consuming, especially with large spatial dimensionality. To address this, we provide various approximations to reduce the complexity, which are detailed in Section 4.4. For all later experiments, except the translational toy example, we use the Batch-Optimal approximation with windows size $b$ 512, with the batch referring to batches of semantic concept vectors $\mathbf{v}$ and not samples. The pseudo-code for calculating *semantic RSMs* is visualized in the Appendix under Algorithm 1.

## 4    Experiments: Semantic vs Spatio-Semantic RSMs

Given the novel permutation-invariant similarity definition, we evaluate the utility of our semantic RSMs relative to spatio-semantic RSMs for various similarity kernels, architectures, and tasks. Across all experiments we compare the linear kernel, the radial basis function (RBF) kernel, and the cosine similarity kernel, see Appendix A for details.

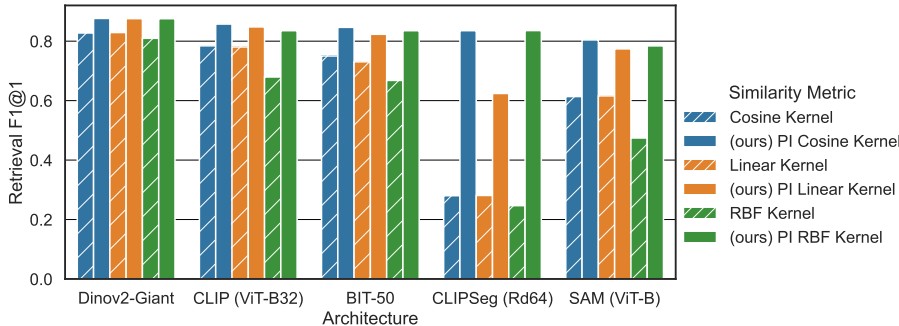

Figure 3: **Relaxing the constraint of spatial alignment leads to better retrieval.** We leverage general feature extractors to embed images of the EgoObjects dataset. We then compare these embeddings either with or without permutation invariance. PI: Permutation Invariant

## 4.1 Translation sensitivity

To illustrate the problems of coupling semantic content and localization a toy dataset is created using $84 \times 84$ pixels large, downsampled images of ImageNet [2]. For each image two $64 \times 64$ crops are extracted, one from the upper-left and one from the lower-left corner, resulting in two images that share $44 \times 64$ identical pixels (Fig. 2 left). Ten upper-left and ten lower-left crops are then used to extract representations of a ResNet18 [7], which are subsequently used to calculate spatio-semantic and semantic RSMs at different layers of the architecture (Fig. 2 middle). As kernel, we use the radial basis function, as it provides bounded similarity values allowing a better visualization.

As expected, the *spatio-semantic* RSM measures low similarity between pairs of overlapping crops, due to the semantic concept vectors not aligning. Only in the last layer, after many pooling operations, the off-diagonal is slightly expressed. Conversely, our *semantic* RSM is capable of detecting the high semantic similarity of the partially overlapping crops throughout the entire depth of the architecture, as evident by the highly similar off-diagonal.

Aggregating the similarity values between partially-overlapping and between different images across multiple batches, allows us to measure the distribution of similarity values between overlapping crops, and non-related image comparisons. Throughout the entire depth of the architecture, the similarity distributions show that our measure better separates overlapping images from different images. Notably, the similarity distribution in *spatio-semantic* RSMs shows a significant overlap of the distributions of partially overlapping images and non-related images, making differentiation between them difficult (Fig. 2 right). A similar toy experiment for a ViT-B/16 [4], is provided in Appendix B.

## 4.2 Similarity-based retrieval

To test the impact of the *semantic* RSMs in real-world applications, we now investigate the common task of image retrieval. Each entry in an RSM quantifies a sample-to-sample similarity value, which can be directly used for retrieval. While not specifically designed for it, we argue that better retrieval performance reflects a better inter-sample similarity. This allows us to quantify improvements in the RSM structure. To measure retrieval performance the EgoObjects dataset [35] is used. It contains frames of video that capture the same scene from different viewing perspectives and lighting conditions. This results in object centers being distributed across the extent of the image.

By randomly sampling 2000 query images and 5000 database images from the test set and using general feature extractors to extract embeddings from them we construct RSMs that allow us to do retrieval. As feature extractors we use CLIP (ViT/B32) [24], CLIPSeg (Rd64) [16], DinoV2-Giant [22], SAM (ViT/B32) [9] and BIT-50 [11] and as kernels for similarity calculation we use the cosine similarity, RBF and the inner product.

For all RSMs, we retrieve the most similar image that is not part of the same video – the same scene but different conditions are allowed. As multiple objects can be present in each scene,

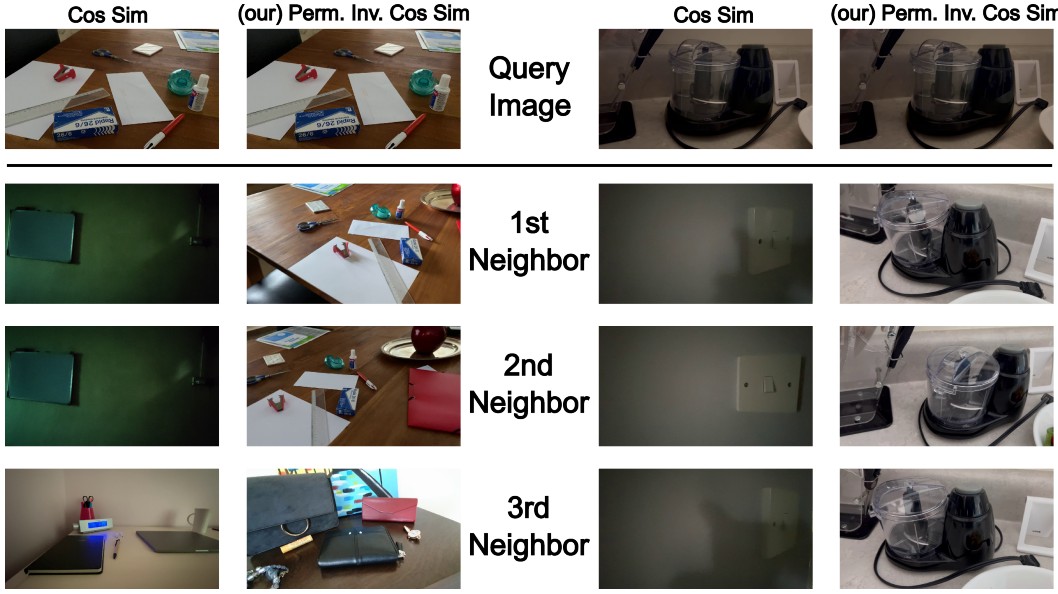

Figure 4: **Retrieving by permutation invariant similarity returns similar scenes of different spatial geometry.** We visualize the top 3 most similar images according to two exemplary query images for SAM ViT/B32.

we quantify retrieval performance by the F1-Score, measuring the overlap of annotated objects between images. Due to the rather complex dataset, we elaborate this in more detail in Appendix D.1.

Across all architectures and metrics, the inclusion of permutation invariance (PI) for the similarity calculation improves retrieval performance relative to the non-invariant similarity, in some cases with a dramatic difference in performance, see Fig. 3. For models designed for dense downstream tasks like SAM or CLIPSeg, the retrieval performance changes particularly much, while models with more global reasoning, like BiT improve less, relatively.

**Qualitative Similarity**   Aside from a quantitative comparison, we visualize the most similar retrieved images for two exemplary queries of SAM in Fig. 4 as case examples.
**Left Query**: The image displays various utensils scattered on a desk. When retrieving with the permutation-invariant similarity metric two images of the same scene but a very different perspective are successfully retrieved as most similar. Retrieving with the non-invariant similarity metric fails to retrieve similar images, due to lack of spatial alignment of the semantic concepts. Instead, it retrieves images of a whiteboard, possibly due to its spatial alignment with the paper on the desk.
**Right Query**: The image features a blender on a counter. The retrieval based on non-permutation-invariant similarity fails to retrieve any of the semantically similar scenes and returns images with a light switch, likely due to the spatial alignment of the light-switch-looking object to the right of the blender. Contrary, the retrieval based on permutation-invariant similarity correctly returns the blender in all cases from different perspectives. Additional qualitative examples are provided in Appendix D.3.

These experiments display clearly, that demanding spatial alignment can be a significant shortcoming when semantically similar concepts are misaligned. In Fig. 4, the network learned to represent the objects very similarly, despite a shift in perspective, but due to the same objects not aligning anymore, spatio-semantic similarity fails to recognize this. This effect should generalize to other datasets where objects are not heavily centered. For datasets with heavy object-centric behavior, like ImageNet, this should be less pronounced.

Table 1: **Similarity invariant to spatial permutations is better at predicting if the class probabilities will be similar.** PI: Permutation Invariant

| Architectures | Pearson Correlation $\rho$ | | | | | |
| | Cosine Sim. | | Inner Product | | RBF | |
| | - | (ours) PI | - | (ours) PI | - | (ours) PI |
| --- | --- | --- | --- | --- | --- | --- |
| ResNet18 | -0.276 | **-0.326** | -0.259 | **-0.270** | -0.176 | **-0.199** |
| ResNet50 | -0.248 | **-0.291** | -0.243 | **-0.261** | 0.040 | **0.029** |
| ResNet101 | -0.192 | **-0.276** | -0.174 | **-0.240** | 0.091 | **0.084** |
| ConvNextV2-Base | **-0.134** | -0.098 | -0.132 | **-0.171** | 0.117 | **0.090** |
| ViT-B/16 | -0.046 | **-0.100** | **-0.045** | -0.026 | -0.077 | **-0.122** |
| ViT-L/32 | -0.138 | **-0.188** | -0.138 | **-0.144** | -0.134 | **-0.166** |
| DinoV2-Giant | -0.012 | **-0.044** | -0.013 | **-0.031** | -0.008 | **-0.048** |

## 4.3 Output similarity vs Representational Similarity

While the retrieval experiments relate to a rather human notion of similarity, one can raise the question if semantic RSMs are also better at measuring the similarity for classifiers.
For each pair of samples, we can compare how similar the predicted class probabilities of a model are and compare this to the representational similarity. A commonly used metric for this is the Jensen-Shannon Divergence (JSD), which quantifies how dissimilar the two probability distributions are from one another. More details are provided in Appendix E.

Consequently, we use various classifiers trained to predict ImageNet1k from Huggingface and compare the Pearson correlation $\rho$ between their JSD and the representational similarity of their last hidden layer. We chose to use the Pearson correlation, as it allows observing a direct linear behavior between representational similarity and predictive similarity. Again we measure semantic similarity and spatio-semantic similarity with different kernels. Due to JSD measuring dissimilarity, we want the correlation to be as negative as possible. As models we use multiple ResNets [7], ViTs [4] , a fine-tuned DinoV2 [22] classifier from and a convnextv2[33] classifier.

The results, displayed in Table 1, show that for almost all architectures and kernels tested, the permutation invariant similarities are better at capturing the notion of what a classifier deems similar. While better than the spatio-semantic similarity, overall correlations are generally low, indicating that either, the similarity metric is confounded by irrelevant representations, or that the kernels should be improved. Moreover, the RBF kernel sometimes provides a positive correlation indicating it is unsuitable to predict the similarity of output probabilities, whereas the Cosine Similarity and the Inner Product both are consistently negative for all architectures tested.

## 4.4 Optimizing runtime

Since we find the best possible permutation matrix through linear sum assignment algorithms that maximize the inner product of two samples, we can guarantee that the $K_{ij,semantic} \geq K_{ij} \forall i, j$. This provides us with an upper bound of similarity that can be leveraged to measure how much of the maximally achievable semantic similarity was measured by the spatio-semantic similarity. Additionally, it can be used as a baseline to estimate the quality of permutation matrices $\mathbf{P}_{ij}$ provided by faster, approximative assignment algorithms.

**Decreasing computational complexity** Determining the optimal permutation between samples poses a substantial computational challenge with a complexity of $\mathcal{O}(S^3)$ for each of the $\mathcal{O}(N^2)$ pairs in the same mini-batch, particularly for early layers with large spatial resolution $S$. Although, in theory, the calculation of the $K$ matrix needs to be conducted only once for the desired representations, applying the method to representations with larger spatial extents becomes impractical with the demands of optimal matching.
To mitigate runtime, two options are available: reducing the batch size $N$ to lessen the number of permutation calculations or decreasing the time spent on finding the permutation. Given that scenarios like image retrieval often desire larger batches, our focus is on minimizing the time required to obtain suitable assignments.

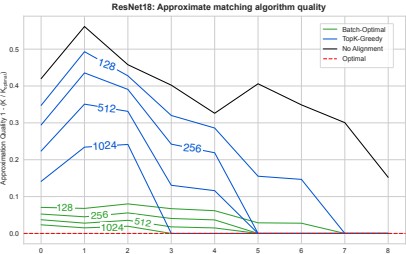
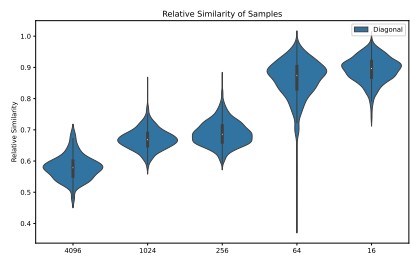

Figure 5: **Approximative algorithms yield comparable matching quality to optimal algorithms.** The ratio of similarity from various approximations relative to maximal semantic similarity is visualized across multiple layers of a ResNet18.

Figure 6: **Relative similarity is not isotropic.** When aligning semantic concepts we observe that similarity changes heterogeneously, indicating that some pairs of samples have more spatially misaligned semantic concepts than others.

Solving the optimal bipartite matching between semantic concept vectors is equivalent to the well-known assignment problem [18, 1]. We attempted to find existing approximate algorithms for this purpose. Unfortunately, most established algorithms primarily focus on optimal solutions, and existing approximate algorithm implementations, such as those based on the auction algorithm [6], are not runtime-optimized, often taking longer than optimal algorithms in our experiments. To enhance computational efficiency nonetheless, we explored three tailored approximation algorithms:

A) A Greedy breadth-first matching (**Greedy**)

B) An optimal matching of the TopK values based on their Norm, followed by the Greedy algorithm for the remaining samples (**TopK-Greedy**)

C) Optimal matching of smaller batches, with samples batched by their Norm (**Batch-Optimal**)

For explicit details on the approximation algorithms, we refer to Appendix F.

We conducted a comprehensive comparison between the approximate algorithms and the optimal algorithm. We compare their runtime per sample and the quality of matches, quantified by the average relative similarity $\frac{k}{k_{optimal}}$. The evaluation utilized representations from a ResNet18 on TinyImageNet, as illustrated in Fig. 5.

It can be seen that the measured spatio-semantic similarity for TinyImageNet samples are, on average, 30% lower with layers of higher spatial resolution exceeding 40%. This suggests a notable misalignment of semantic concept vectors. Notably, the Batch-Optimal approximation stands out as a reliable approximation for optimal matching. The fastest of the Batch-Optimal approximation methods shows $< 8\%$ error while improving run-time $\times 36$ relative to the fastest optimal algorithm for spatial extent $4096$, while no spatial alignment shows $42\%$ deviation from the optimal matching. Moreover, we highlight the time vs accuracy trade-off of the different optimal and approximate algorithms in Appendix F.1. Furthermore, it can be seen that the changes between spatio-semantic and semantic RSMs are anisotropic, as highlighted in Fig. 6, indicating scale invariant downstream applications may be influenced.

## 5   Discussion, Limitations, and Conclusion

The concept of Representational Similarity Matrices (RSMs) is a powerful tool to represent the similarity structure of complex systems. In this paper we revisit the construction of such RSMs for neural networks of the vision domain, question the current state, and propose *semantic RSMs*, warranting discussion.

**Spatio-semantic coupling**  Being aware that current, *spatio-semantic* RSMs demand semantic concepts to be aligned is highly relevant to understand what RSMs are sensitive to. Previous work [32] identified this shortcoming and proposed translation invariance, partially addressing this issue. We argue translation invariance is insufficient, since models may learn invariances during training, which the translation invariant metric would not be sensitive to. Subsequently, we propose a new – spatially permutation invariant – similarity measure between samples that allows the detection of similarity whenever a model expresses similar semantic vectors in its representations, irrespective of spatial geometry. To highlight the benefits of our similarity, we propose that better similarity measures should allow more accurate retrieval when comparing last-layer representations and should allow better predictions about the similarity of class probabilities of a classifier. However, we acknowledge certain limitations in our current evaluation. Specifically, we have not yet compared our method to more established retrieval techniques. Traditional retrieval methods are often not applied to representations directly but utilize a lower-dimensional non-spatial, global vector representing the entire sample. In contrast, we chose to limit ourselves to methods that are directly applied to the representations.

**Computational Complexity**  Aside from quantitative or qualitative benefits, the construction of semantic RSMs is time-consuming, limiting its applicability. This complexity mostly affects layers of large spatial extent, which mostly corresponds to early CNN layers while later layers and ViTs are unproblematic. Our proposed Batch-optimal approximation alleviates this partially, yet application to large-scale representations at higher resolution, like at the output of a segmentation architecture with spatial extents of s=65.536 would be too costly. We leave optimizing the compute efficiency or finding better approximations for future work.

**Conclusion**  In conclusion, our investigation into semantic RSMs has shed light on the limitations of spatio-semantic RSMs and introduced a novel approach to disentangle spatial alignment from semantic similarity. The proposed method provides a more accurate measure of how representations capture underlying semantic content, showcasing its potential in various applications, particularly in scenarios where spatial alignment cannot be assumed. While challenges such as computational complexity and scalability need to be addressed, the findings open avenues for further research and improvement in the analysis of neural network representations.

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

# A  Kernel function definitions

Defining the notion of similarity between two vectors is a matter of preference and wanted properties. In this work, we include the Radial Basis Function kernel Eq. (6) and the linear kernel Eq. (7), as well as the cosine similarity kernel Eq. (8). We include the former two as they were proposed for RSM construction by Kornblith et al. [12] and the cosine similarity due to its popularity in the retrieval domain, despite generally being applied to class probabilities.

We denote that in our manuscript we refer to the linear kernel as the inner product and dot product interchangeably as they correspond to the same operation.

$$k_{\text{RBF}}(\mathbf{x}, \mathbf{y}) = \exp\left(-\frac{\|\mathbf{x} - \mathbf{y}\|^2}{2\sigma^2}\right) \tag{6}$$

$$k_{\text{linear}}(\mathbf{x}, \mathbf{y}) = \langle \mathbf{x}, \mathbf{y} \rangle \tag{7}$$

$$k_{\text{cosine}}(\mathbf{x}, \mathbf{y}) = \frac{\langle \mathbf{x}, \mathbf{y} \rangle}{\|\mathbf{x}\|\|\mathbf{y}\|} \tag{8}$$

**Different properties**  The three selected kernels all have different properties: The RBF kernel and the Cosine similarity are bounded between $k_{RBF}(x, y), k_{cosine}(x, y) \in [0, 1] \quad \forall x, y \in \mathcal{R}$, while the $k_{linear}(x, y)$ is not bounded.
Moreover, the RBF kernel is parametrized by $\sigma$, which influences at which rate the distance between the representations results in a decrease in similarity. For all our experiments we choose $\sigma$ as the square root of the median Euclidean distance of all distances within a mini-batch.

# B  Translation Sensitivity of a ViT-B/16

Similarly to the translated Tiny-ImageNet experiment, we prepared a very similar experiment for a ViT-B/16 vision transformer that was pre-trained on ImageNet1k. We utilize the implementation and weights provided by torchvision [17].
Given the larger ImageNet images, we resample them to $324 \times 324$ pixels and crop two partially overlapping images of size $224 \times 224$ from it. Given these image pairs, we calculate semantic and spatio-semantic RSMs again, see Fig. 7.
It is important to note that our approach intentionally avoids achieving a perfect translation that would lead to the same patchified tokens, as we shift the image by a factor that is not divisible by 16, the patching window size. We believe this realistic imperfection is preferable to a perfect overlap, where identical tokens would be formed from the exact same set of pixels.
Similarly to the previous partially overlapping crop experiment, we can observe that spatio-semantic RSMs are incapable of identifying the largely identical content of the two partially overlapping crops due to their different localization. The spatio-semantic RSMs can capture this notion of similarity as evidenced by the off-diagonal and the larger gap in the distribution of similarity between partially overlapping samples and independent samples, Fig. 7 (bottom).
Contrary to the CNN example in the main, overall similarity between samples is much lower overall and separation between translated image pairs and random image pairs follows a vastly different trajectory. While there is a profound difference, the origin of this difference cannot be clearly made out. We hypothesize that this may be due to the different ways that Transformers process information and learn different representations, as highlighted in Raghu et al. [26].

Moreover, we emphasize, that calculating the optimal permutation is significantly faster for ViTs than the CNNs, as the early tokenization reduces the spatial dimension $S$ substantially at an early stage, whereas the iterative downsampling of CNNs makes comparing representations of early layers very costly.

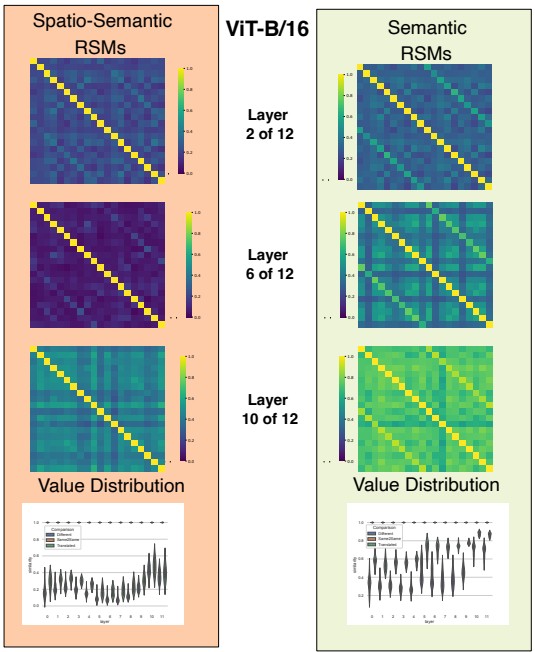

Figure 7: **Semantic RSMs do also capture spatial translations for token sequences of ViT's**. We calculate spatio-semantic and semantic RSMs with the Radial Basis Function (RBF) kernel for a ViT-B/16 with representations extracted from ImageNet. Similarly, we introduce partially-overlapping crops that get tokenized and processed by the ViT. Due to the initial shift, the token sequence does not align anymore between the crops. Similar to the CNNs, spatio-semantic RSMs exhibit low similarity values in the off-diagonals, providing a limited indication of overlapping content between crops. In contrast, Semantic RSMs prove notably more effective in discerning substantial overlap, offering higher similarity values in the off-diagonals and thereby indicating a greater degree of similarity between large portions of the image.

## C   Pseudo Code

In addition to our provided explanation of the algorithm in the main manuscript, we provide the pseudo-code used to compute semantic RSMs in Algorithm 1. The only difference between semantic and spatio-semantic RSMs algorithmically is where the optimal permutation is calculated that maximally aligns the two representations. The current definition of spatio-semantic RSMs assumes that spatial locations are corresponding, while semantic RSMs calculate correspondence through similarity matching.

## D   Additions: Retrieval Experiment

In addition to the provided retrieval examples in the main manuscript, we provide more details on the retrieval experiments in Appendix D.1, an additional table holding the quantitative data of Fig. 3 with varying database sizes in Appendix D.2 and lastly, additional qualitative retrieval examples for each model including direct comparisons of all models for the same query image in Appendix D.3.

### D.1   Details of Retrieval Experiment

For the retrieval experiment, we utilize the EgoObjects dataset [35]. It contains multiple frames from multiple videos, with multiple videos capturing the same scene under different shifts like lighting conditions, distances, viewing angles, and different motion trajectories. For each frame, multiple objects of different categories can be present and are annotated

**Algorithm 1: Semantic RSM calculation.** We calculate the optimal permutation matrix, resulting in maximal similarity between the representations of two samples.

---

**Data:** $Z \in \mathbb{R}^{N \times C \times S}$; Kernel $k$
**Result:** Semantic RSM $K$
$\mathbf{K} = \mathbf{0} \in \mathbb{R}^{N \times N}$
**for** $i$ *from 0 to N* **do**
    **for** $j$ *from 0 to N* **do**
        **if** $i < j$ **then**
             cont.
        **if** $i \neq j$ **then**
             $\mathbf{P}_{ij} = argmax_{\mathbf{P}_{ij}} \langle z_i, \mathbf{P}_{ij} z_j \rangle$;
             $z'_j = \mathbf{P}_{ij} z_j$;
        **else**
             $z'_j = z_j$;
        $K_{ij} = k(z_i, z'_j)$;
        $K_{ji} = K_{ij}$ ;                                 /* is symmetric */

---

through a bounding box. Moreover, frames can vary in spatial resolution, yet a large fraction was captured in 16:9 format of $1920 \times 1028$ pixels.

**Image preprocessing** For our experiments, we utilize the EgoObjects test set, which is comprised of 29.5K images. Of these 29.5k we remove all images not in 16:9 format and resize the remaining to the $1920 \times 1028$ format. This discards roughly 10k images.

Of all remaining images, we then draw 2k query images and 5k database images used for extracting embeddings for similarity calculation and later retrieval. Naturally, we sample in a way to keep the 2k query and 5k database image sets non-overlapping. When passing the images to the models for feature extraction each image is preprocessed according to the corresponding Huggingface ImagePreprocessor. This mostly represents resizing the image by the shortest edge to the expected image input dimensions and normalizing the image. The only exception is SAM, of which we use the official implementation, which handles feature extraction and embedding of the image itself.

**Feature Extraction and preparation** As mentioned in the main manuscript we use

1. CLIP (ViT/B32) [24]
2. ClipSeg (Rd64) [16]
3. DinoV2-Giant [22]
4. SAM (ViT/B32) [9] and
5. BIT-50 [11]

as general feature extractors as they were trained on a vast amount of data.[4].

Of all models, we use the last hidden layer as image embeddings should they not per-default provide image embeddings as output. After extracting representations we calculate the mean from the database embeddings to zero-center all representations by, query and database representations alike.

**RSM construction** Given all 2000 query embeddings and 5000 database embeddings, we calculate the RSMs. To parallelize this process we mini-batch the representations into $100 \times 100$ pairs and populate the $2000 \times 5000$ matrix in this fashion. We denote that this proved to be necessary for models with large spatial embedding dimensions like SAM, starring $64 \times 64$ spatial extent. Moreover, we denote that, due to the RBF choosing its

---

[4]Last accessed on 22nd of May 2024

parameter based on the median of the measured values within one batch this patch-wise calculation is not optimal for this kernel. The inner product and the cosine similarity kernels are not affected by this.

**Retrieval measurement**   Given the RSMs containing a sample-to-sample similarity measure, we can retrieve the most similar sample of the database for each query from it.

As each image can contain multiple objects measuring retrieval performance is not trivial. For both images we quantify how many objects of each class are present in the image, resulting in a count of class instances for each image. With the query image representing the ground truth (GT) and the database representing the prediction, we match class instance counts. Each correctly matched GT instance represents a TP, each missed an FN and all unmatched database instances represent an FP.

To formalize: Let $Q_c$ be the number of instances of class $c$ in the query image and $D_c$ be the number of instances of class $c$ in the database image. With this, the used F1 metric can be expressed as

$$TP = \sum_{c \in C} \min(Q_c, D_c) \tag{9}$$

$$FN = \sum_{c \in C} \max(0, Q_c - D_c) \tag{10}$$

$$FP = \sum_{c \in C} FP_c = \sum_{c \in C} \max(0, D_c - Q_c) \tag{11}$$

$$F1 = \frac{2 \cdot TP}{2 \cdot TP + FP + FN} \tag{12}$$

## D.2   Additional Quantitative Retrieval data

In addition to the results highlighted in Fig. 3 we provide retrieval results for varying database sizes to retrieve from EgoObjects. Specifically, results for database sizes of 2.5k, 5k, and 10k samples are given in Table 2.

Table 2: Quantitative results of the EgoObject retrieval experiment for multiple models and multiple database sizes. Database Size 5.000 represents Fig. 3 of the main manuscript. It can be observed that models with a greater spatial extent (CLIPSeg and SAM) show greater improvement in retrieval performance over models with a lower spatial extent (DinoV2-Giant, BiT-50, CLIP). Moreover, it can be observed that retrieval improvements decrease with growing database size. This is likely due to the increasing odds of finding one image where spatial position and semantics align w.r.t. the query images. PI: Permutation Invariant, Diff: Difference (PI - None)

| Database Size (N) | | 2.500 | | | 5.000 | | | 10.000 | | |
|---|---|---|---|---|---|---|---|---|---|---|
| Invariance | | None | PI | Diff | None | PI | Diff | None | PI | Diff |
| Architecture | Kernel | | F1@1 | | | F1@1 | | | F1@1 | |
| | Cosine Sim. | 0.240 | 0.781 | 0.541 | 0.280 | 0.835 | 0.555 | 0.344 | 0.859 | 0.515 |
| CLIPSeg | Inner Product | 0.233 | 0.548 | 0.315 | 0.281 | 0.624 | 0.343 | 0.328 | 0.569 | 0.241 |
| | RBF | 0.205 | 0.781 | 0.576 | 0.247 | 0.836 | 0.589 | 0.293 | 0.854 | 0.561 |
| | Cosine Sim. | 0.765 | 0.840 | 0.075 | 0.827 | 0.876 | 0.049 | 0.857 | 0.889 | 0.032 |
| DinoV2-Giant | Inner Product | 0.767 | 0.839 | 0.072 | 0.829 | 0.876 | 0.047 | 0.856 | 0.890 | 0.033 |
| | RBF | 0.735 | 0.834 | 0.099 | 0.810 | 0.876 | 0.066 | 0.843 | 0.888 | 0.045 |
| | Cosine Sim. | 0.670 | 0.799 | 0.129 | 0.750 | 0.846 | 0.096 | 0.800 | 0.868 | 0.068 |
| BiT-50 | Inner Product | 0.659 | 0.780 | 0.122 | 0.730 | 0.823 | 0.093 | 0.782 | 0.852 | 0.069 |
| | RBF | 0.582 | 0.789 | 0.208 | 0.668 | 0.835 | 0.167 | 0.741 | 0.859 | 0.118 |
| | Cosine Sim. | 0.704 | 0.803 | 0.100 | 0.784 | 0.858 | 0.073 | 0.827 | 0.876 | 0.049 |
| CLIP | Inner Product | 0.703 | 0.792 | 0.090 | 0.780 | 0.848 | 0.067 | 0.823 | 0.867 | 0.044 |
| | RBF | 0.584 | 0.779 | 0.196 | 0.679 | 0.835 | 0.156 | 0.749 | 0.863 | 0.114 |
| | Cosine Sim. | 0.509 | 0.735 | 0.226 | 0.614 | 0.804 | 0.190 | 0.688 | 0.838 | 0.151 |
| SAM ViT/B32 | Inner | 0.511 | 0.695 | 0.183 | 0.616 | 0.774 | 0.158 | 0.688 | 0.815 | 0.126 |
| | RBF | 0.371 | 0.703 | 0.332 | 0.474 | 0.784 | 0.310 | 0.556 | 0.823 | 0.267 |

## D.3 Additional Qualitative Examples

In addition to the two qualitative examples provided for SAM in the main, additional examples are provided. Qualitative examples are picked from the first 50 query images when retrieving from a database size of 5.000 images. Additional qualitative retrieval examples are provided for DinoV2 in Fig. 8, CLIP in Fig. 9, BiT-50 in Fig. 10, CLIPSeg in Fig. 11 and SAM in Fig. 12. Moreover, we highlight a direct comparison between all models for the same images in Fig. 13 and Fig. 14.

Across all models, it can be observed that models retrieve images where semantic content and spatial positions are aligned when using standard cosine similarity. This leads to larger-scale objects like desks, tiled floors, or countertops dominating retrieval when imaged from a similar perspective. When decoupling spatial-alignment from semantic content, images of the same scene but a different perspective get retrieved more often, leading to the same scene appearing more regularly in the 5 nearest neighbors. This effect is currently not quantified in the F1 metric, due to only comparing object presence between the query and the 1st neighbor and not the average F1 between the query and the top 5 neighbors. We opted against using this, as retrieval metrics most commonly consider the maximum match in the top 5 neighbors.

Moreover, it can be observed that models with lower spatial extent can retrieve quite different neighbors as opposed to models with higher spatial extent, see Fig. 13. While DinoV2 and CLIP retrieve very similar 4th and 5th neighbors of the same book object without the smaller mouse and headphone object, CLIPSeg and SAM retrieve scenes with these two objects still present instead.

## D.4 Cityscapes Quantitative Retrieval

Since previous results were limited to the EgoObjects dataset we provide an additional quantitative experiment on CityScapes. Analog to before, we use N=500 validation images as the query dataset and the remaining N=2975 training images as the database for retrieval. We utilize the IoU metric to compare the presence of semantic classes between the images.

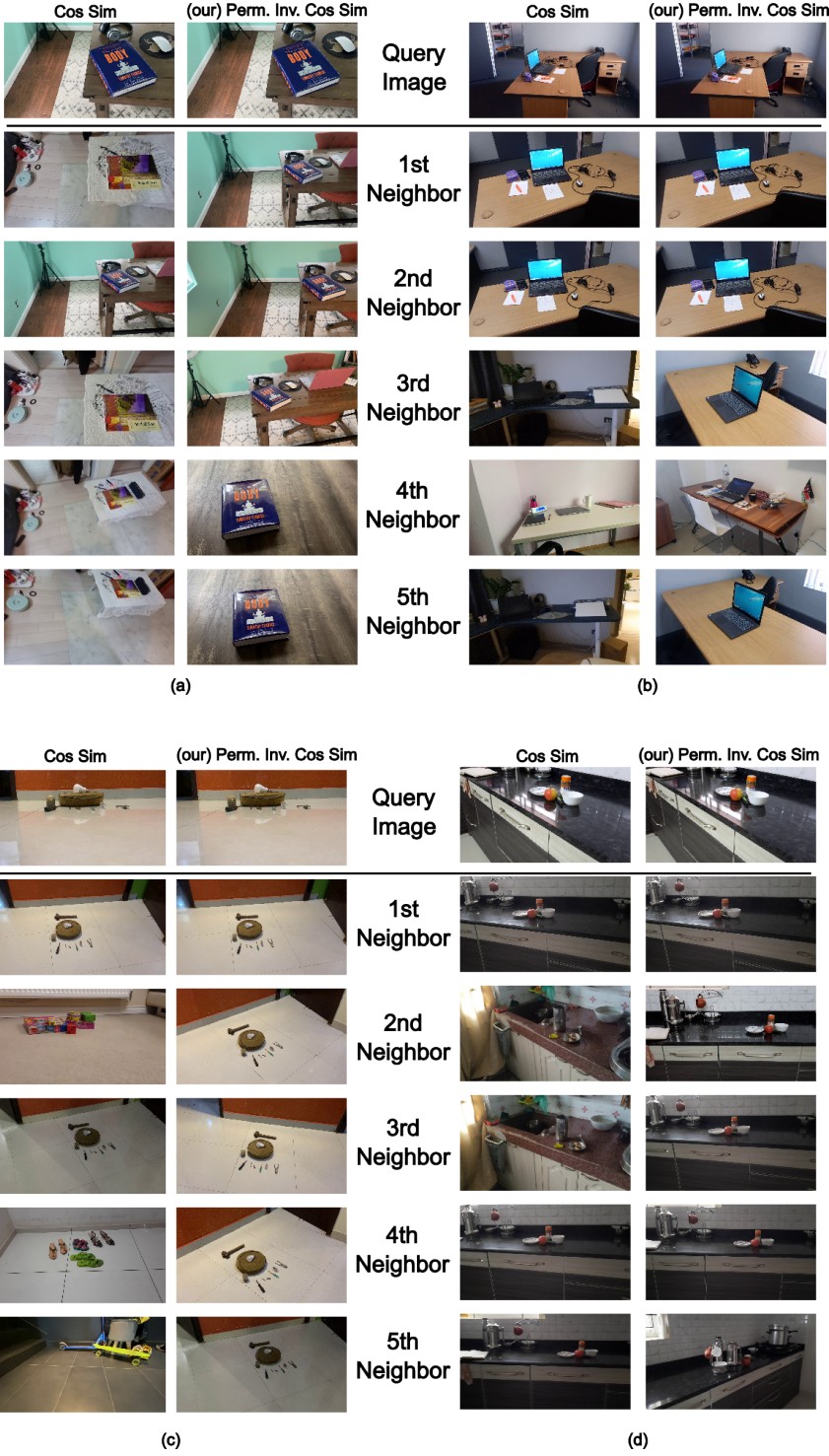

Figure 8: **Additional qualitative retrieval samples for DinoV2.** We visualize the top 5 most similar neighbors for four query images.

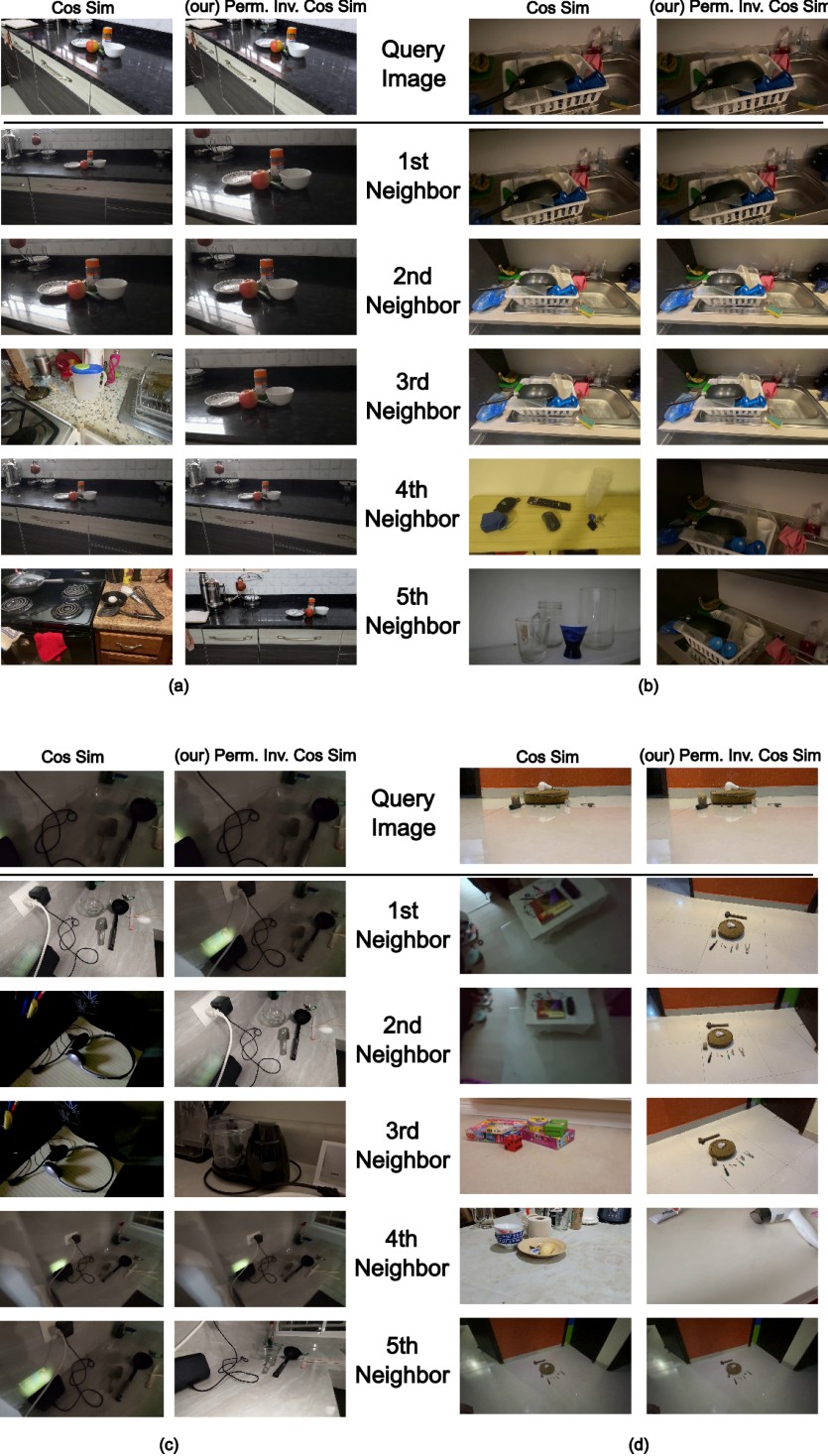

Figure 9: **Additional qualitative retrieval samples for CLIP.** We visualize the top 5 most similar neighbors for four query images.

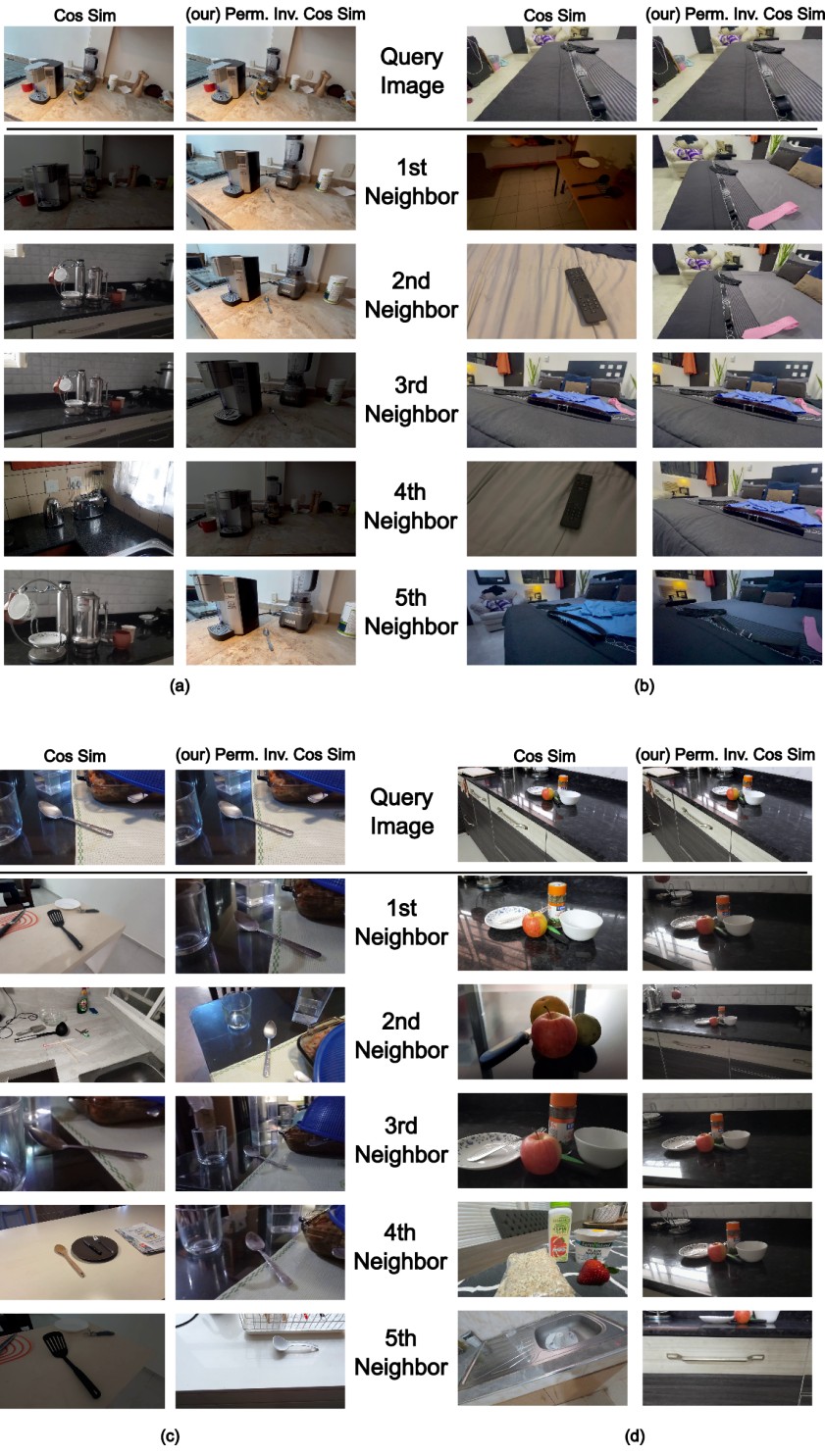

Figure 10: **Additional qualitative retrieval samples for BiT-50.** We visualize the top 5 most similar neighbors for four query images.

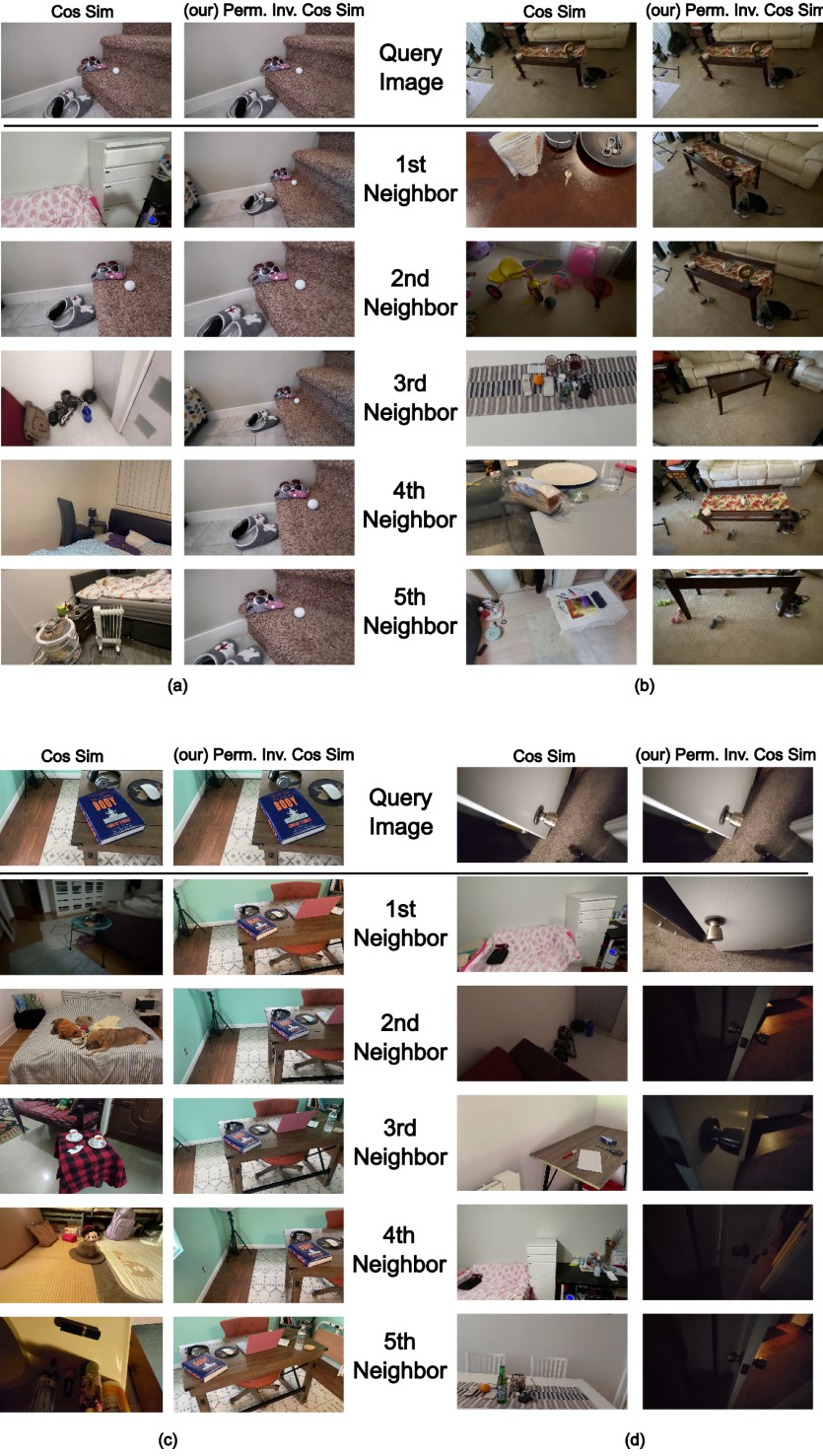

Figure 11: **Additional qualitative retrieval samples for CLIPSeg.** We visualize the top 5 most similar neighbors for four query images.

# Qualitative Examples of SAM

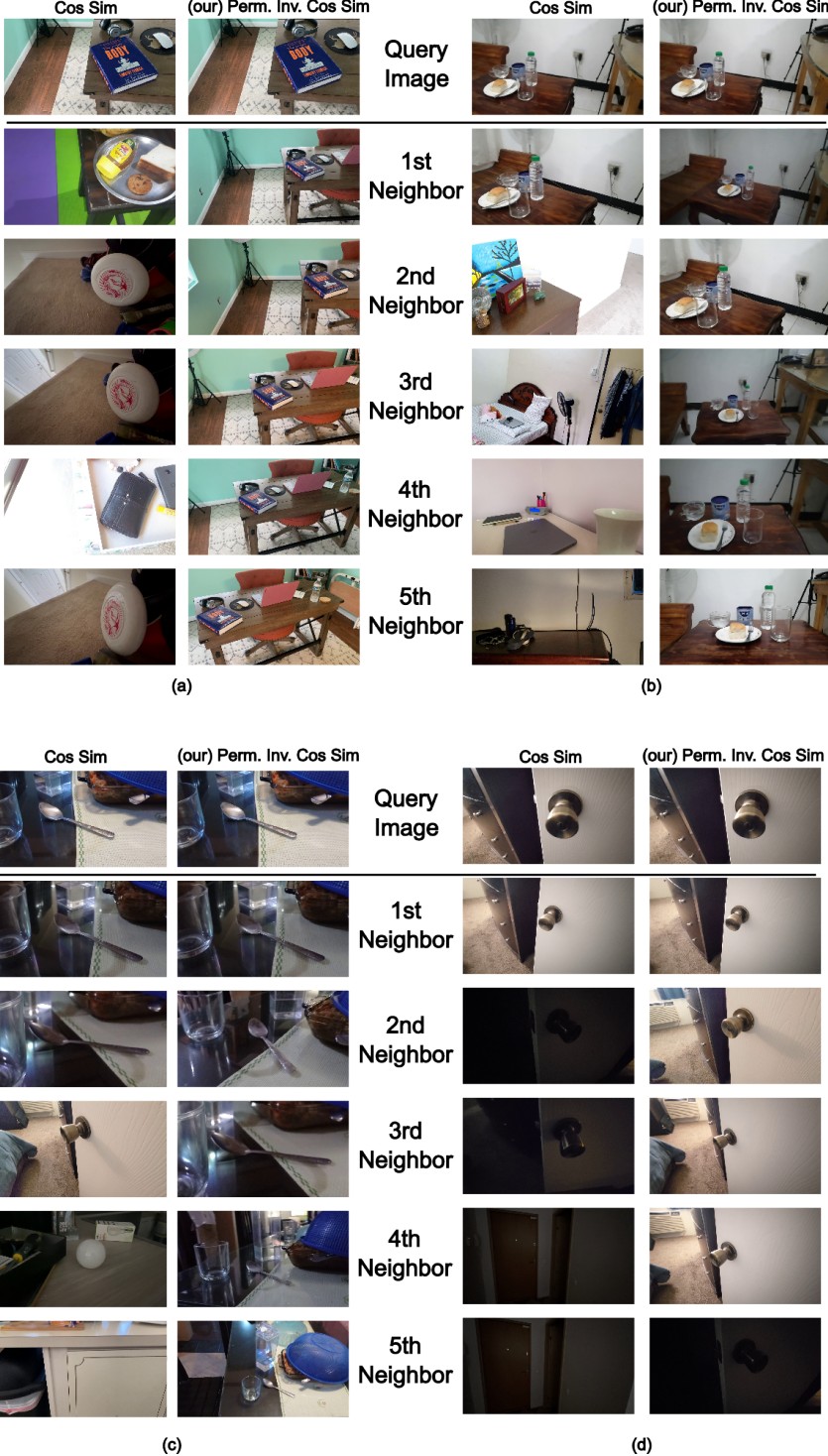

Figure 12: **Additional qualitative retrieval samples for SAM.** We visualize the top 5 most similar neighbors for four query images.

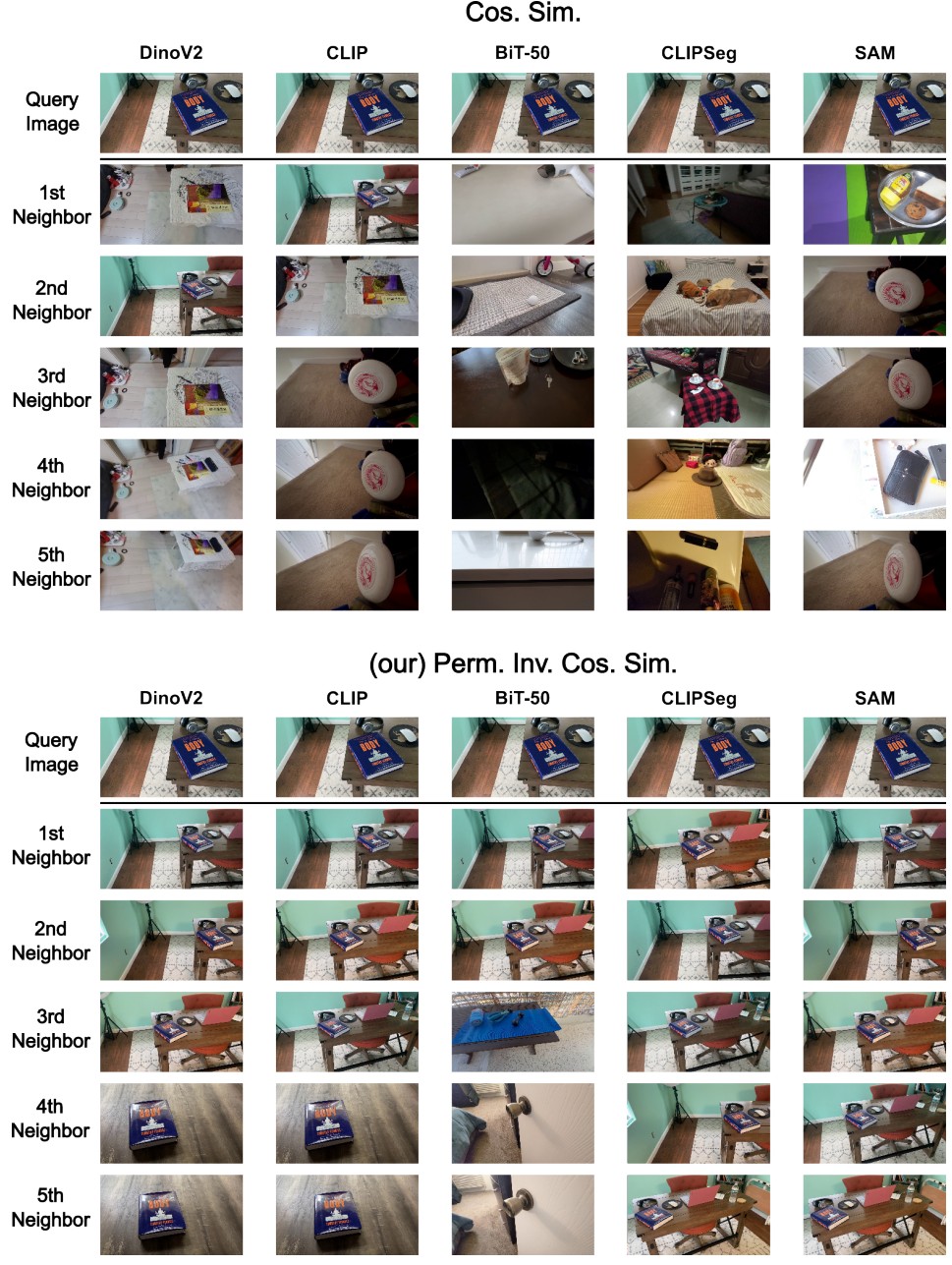

Figure 13: **Direct comparison of models.** We visualize the top 5 most similar images of all models retrieved through cosine similarity or permutation invariant cosine similarity.

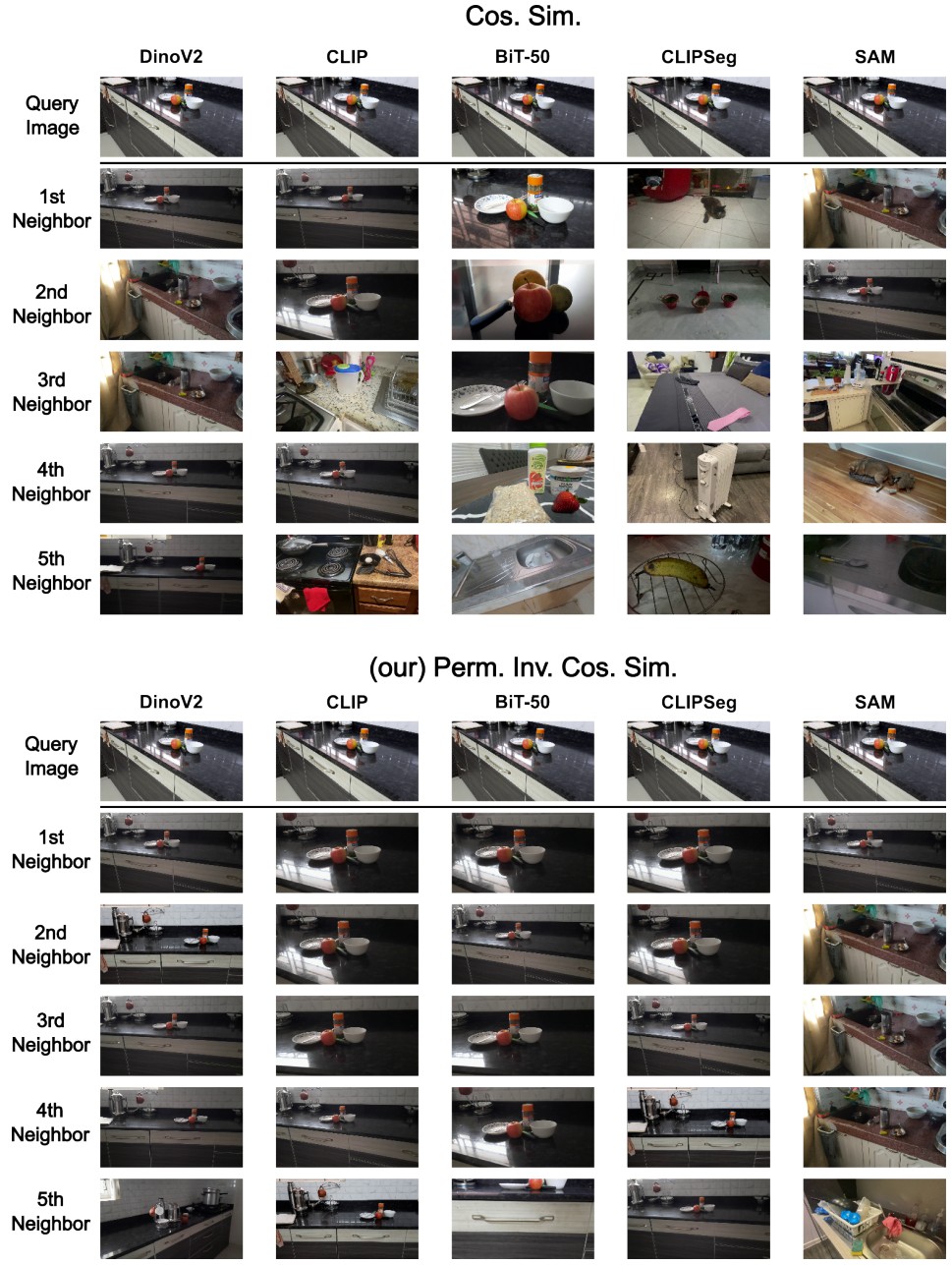

Figure 14: **Direct comparison of models.** We visualize the top 5 most similar images of all models retrieved through cosine similarity or permutation invariant cosine similarity.

Due to the lack of instance labels in the CityScapes dataset, we can't take the quantity of objects into account as before, but can only consider if a semantic class is present or absent. As apparent from Table 3 permutation invariance allows for consistently improved retrieval when utilizing cosine similarity or RBF kernels.

Table 3: Retrieval results for the Cityscapes Dataset. We retrieve the most similar image according to RSMs and calculate the IoU of query semantic classes and retrieved semantic classes. Overall query images used are validation images of size N=500 and the database are the training images of size N=2975. Differences between metrics are low, due to many images containing a large number of classes and the lack of instance label information. Despite this, permutation invariance improves Cosine Sim and RBF retrieval performance consistently, with the Inner Product showing mixed results.

| Invariance Architectures | Cosine Sim. | | Inner Product | | RBF | |
|---|---|---|---|---|---|---|
| | - | (ours) PI | - | (ours) PI | - | (ours) PI |
| CLIPSeg | 0.662 | **0.694** | **0.664** | 0.654 | 0.664 | **0.696** |
| DinoV2-Giant | 0.689 | **0.700** | 0.686 | **0.696** | 0.686 | **0.690** |
| BiT-50 | 0.679 | **0.687** | 0.677 | 0.677 | 0.684 | **0.687** |
| CLIP | 0.691 | **0.701** | 0.693 | **0.701** | 0.678 | **0.687** |
| SAM ViT/B32 | 0.690 | **0.702** | **0.687** | 0.677 | 0.679 | **0.687** |

# E  Details: Output similarity vs Representational Similarity

To measure the correlation between the inter-sample representational similarity and the prediction probability inter-sample similarity we utilize pre-trained classifiers and the ImageNet1k [2] dataset. Unlike during the retrieval results this constraints the possible model selections to models trained for classification.

**Image preprocessing**  Test set images of ImageNet1k are randomly sampled without applying any filtering to them. In total, we utilize a subset of 2k ImageNet test set samples in this experiment. This may appear small, yet provides a sufficient basis as the combinatoric growth increases the absolute number of measurements substantially.

**Feature and logit extraction and preparation**  As mentioned in the main manuscript we use

1. ResNet18 [7]
2. ResNet50 [7]
3. ResNet101 [7]
4. a DinoV2-Giant based classifier [22]
5. ConvNeXt V2  [33]
6. ViT-B/16  [4] and
7. ViT-L/32 [4] and

as pre-trained classifiers for predicting the ImageNet1k classes.[5]. For each sample, we extract the last hidden layer's representations and center them analog to before. For the same sample, we extract the logits and obtain the probability distribution through the softmax, saving the pair for later comparisons.

**Correlation measurement**  For each pair of representations and probabilities, we calculate the similarities between their representations for all three kernel functions, once permutation invariant and once not. Additionally, we calculate the Jensen-Shannon Divergence (JSD)

---

[5]Last accessed on 22nd of May 2024

between the predicted class probability distributions for $P$ and $Q$. A formal definition of the JSD is provided in Eq. (13).

$$\mathrm{JSD}(P \parallel Q) = \frac{1}{2} D_{\mathrm{KL}}(P \parallel M) + \frac{1}{2} D_{\mathrm{KL}}(Q \parallel M) \tag{13}$$

where $M$ is the pointwise mean of $P$ and $Q$:

$$M = \frac{1}{2}(P + Q) \tag{14}$$

and $D_{\mathrm{KL}}$ is the Kullback-Leibler divergence defined as:

$$D_{\mathrm{KL}}(P \parallel Q) = \sum_i P(i) \log \frac{P(i)}{Q(i)} \tag{15}$$

Given the paired JSD and Similarity $K$ between all samples $i, j$ we utilize the Pearson correlation $\rho$ to calculate the correlation between the two. Due to the JSD being 0 for identical probabilities and increasing for more dissimilar values and the Similarity being 1 for perfectly similar representations and 0 for dissimilar representations, the desired correlation between the two should be negative.

### E.1 Additional correlation results

In addition to the Pearson correlation between the Jensen-Shannon-Divergence (JSD) and inter-image similarity, we also present the results of their relationship measured by the Spearman correlation, as shown in Table 4.
While the Pearson correlation demonstrated consistently stronger correlations with cosine similarity, the inner product, and the RBF kernel, the Spearman rank correlations are less stable across these methods.
For ResNets, we observe a significant decline in correlation consistency and strength with the exception of ResNet18. Opposed to this, ViTs display notably higher negative correlation values when using cosine similarity and radial basis function kernels, in contrast to the Pearson correlation results.

Table 4: Correlation between the representational similarity and the output probabilities of multiple images. In total 20k samples of the IN1k test set are used (as opposed to 2k in the table in the main)

| Metric
Kernel
Invariance
Architecture | Pearson Correlation | | | | | | Spearman's Rank Correlation | | | | | |
|---|---|---|---|---|---|---|---|---|---|---|---|---|
| | Cosine Sim. | | Inner Product | | RBF | | Cosine Sim. | | Inner Product | | RBF | |
| | - | PI | - | PI | - | PI | - | PI | - | PI | - | PI |
| ResNet18 | -0.279 | **-0.328** | -0.264 | **-0.272** | -0.174 | **-0.197** | -0.231 | **-0.337** | **-0.239** | -0.225 | -0.435 | **-0.476** |
| ResNet50 | -0.256 | **-0.305** | -0.249 | **-0.269** | 0.028 | **0.015** | **-0.032** | 0.007 | **-0.046** | -0.040 | 0.128 | 0.132 |
| ResNet101 | -0.235 | **-0.330** | -0.211 | **-0.274** | 0.076 | 0.067 | -0.007 | **-0.053** | -0.007 | **-0.077** | 0.071 | **0.068** |
| ConvNextV2-Base | **-0.162** | -0.126 | -0.160 | **-0.184** | 0.077 | 0.050 | **-0.017** | 0.026 | -0.013 | **-0.045** | 0.143 | **0.128** |
| ViT-B/16 | -0.058 | **-0.098** | **-0.056** | -0.031 | -0.079 | **-0.120** | -0.013 | **-0.230** | **-0.021** | 0.029 | -0.220 | **-0.313** |
| ViT-L/32 | -0.142 | **-0.189** | -0.143 | **-0.152** | -0.131 | **-0.164** | -0.034 | **-0.276** | **-0.029** | -0.014 | -0.335 | **-0.392** |
| DinoV2-Giant | -0.016 | **-0.046** | -0.016 | **-0.030** | -0.013 | **-0.052** | -0.014 | **-0.037** | -0.015 | **-0.022** | -0.015 | **-0.042** |

## F Details of the Approximation Algorithms

The computational complexity of determining optimal matchings using the Jonker-Volgenant algorithm [8] scales significantly with $\mathcal{O}(s^3)$, resulting in substantial computation time for input-patch sizes with spatial dimensions $s = 64^2$. To address this challenge, we propose alternative approximate algorithms with reduced computational complexity. In all our approximations, we take advantage of additional information, specifically the L2-norm $\|v_i\|_2$ of each semantic concept vector. We assume that achieving a high degree of matching involves pairing vectors with the highest norms and high cosine similarity. This assumption guides our design of more efficient matching algorithms.

1. **Greedy:** The simplest approach we employ is breadth-first matching. We determine the order in which to match $v_i$ by considering the L2-norm $\|v_i\|_2$ in descending order. We then match the current $v_i$ with the best, non-assigned $v_j$ based on $A_{ij}$. The sorting complexity is $\mathcal{O}(s \log(s))$, making this the fastest approximate algorithm among those tested.

2. **TopK-Greedy:** Recognizing that the TopK norm concept vectors $\|v_i\|_2$ might have a significantly higher impact on the final similarity, we attempt to find the optimal matching for only the highest TopK norm concept vectors $v_i$ and $v_j$. The remaining lower norm concept vectors are assigned using the Greedy algorithm as described above. The process involves an initial sorting based on the semantic concept vectors' L2-norm, followed by optimal matching with $\mathcal{O}(k^3)$ complexity for the $k$ TopK values and the greedy matching for the remaining values.

3. **Batch-Optimal:** If the TopK norm concept vectors do not sufficiently approximate an optimal matching, we apply optimal matching for the remaining concept vectors in batches. To achieve this, we create $s//b$ smaller batches, with semantic concept vectors assigned to batches according to their L2-norm. All values within a batch are then optimally matched, leading to a matching complexity of $(\frac{s}{b}) \cdot \mathcal{O}(b^3)$.[6]

Evaluating the various approximations, we observe that the Greedy matching yields suboptimal approximation quality and offers marginal to no improvement over the current same-position assignment. Although we do not present the details of the greedy matching, it is important to highlight that it is guaranteed to be worse or equal to the TopK-Greedy matching with a $k$ value of 128, as shown in Fig. 5. We include the Greedy algorithm for completeness as a simple baseline.

Furthermore, it is noteworthy that the TopK-Greedy matching demonstrates that exclusively matching the largest norm concept vectors is insufficient for a good approximation of the optimal matching. This insight suggests that a substantial portion of the overall similarity is contributed by semantic concept vectors not included in the set of highest norms.

Lastly, we observe that the Batch-Optimal approximation, using a small batch size of 128 samples, provides an approximation with less than $10\%$ error compared to the optimal matching. This result underscores the effectiveness of our batching approach based on the L2-norm of the concept vectors. It offers a reliable estimate for overall similarity, simplifying the matching process significantly.

### F.1    Runtime Evaluation

In order to assess algorithm performance across different spatial resolutions, we conducted a benchmarking study. For each resolution, we randomly selected 10,000 pairs of samples from a ResNet101 trained on Tiny-ImageNet. Affinity matrices ($\mathbf{A}ij$) were pre-computed to facilitate permutation ($\mathbf{P}ij$) calculations. The average time taken per matching was then reported for the same single CPU core, as outlined in Table 5. We observe that the OR-Tools implementation outperforms other alternatives, being four times faster than the lapjv implementation [7]. However, even this optimal approach requires 1.52 seconds per pair on a $64 \times 64$ resolution. Despite the potential for parallelizing sample-wise matching, optimal algorithms face scalability challenges with larger spatial dimensions $S$. In contrast, the Batch-Optimal approximation offers a compelling balance between computation time and approximation quality. Importantly, its complexity scales linearly with $S$ due to the fixed batch size.

## G    Semantic RSMs and CKA – Qualitative changes

Building upon the success of Linear/RBF (*spatio-semantic*) CKA to compare systems, we provide some preliminary qualitative comparisons gauging how CKA comparisons are affected by our differently proposed RSM. Unfortunately, hardly any quantitative benchmark

---

[6]There is an error in the current version in the main regarding the square root of $s$, which will be corrected in a revision. We apologize for any confusion

[7]Implementation on Github https://github.com/src-d/lapjv

Table 5: We compare different implementations of the optimal Jonker-Volgenant algorithm [8] against our linearly scaling Batch-Optimal approximation and no matching. The table presents average run-times per pair ($T$) and average similarity relative to the optimal value ($\frac{k}{k_{\text{opt}}}$) for 1,000 randomly chosen image pairs. Utilizing representations of varying sizes from a ResNet101 trained on Tiny-ImageNet, the optimal solutions are reported relative to the maximum similarity achieved by the optimal algorithms. The Batch-Optimal approximation demonstrates a substantial fraction of optimal matching performance with significantly improved scaling.

| Category | Complexity | Matching Algo | Batch $b$ | $s = 256$ | | $s = 1024$ | | $s = 4096$ | |
|---|---|---|---|---|---|---|---|---|---|
| | | | | $\frac{k}{k_{opt}}$ [%] | T | $\frac{k}{k_{opt}}$ [%] | T | $\frac{k}{k_{opt}}$ [%] | T |
| Optimal | $\mathcal{O}(s^3)$ | OR-Tools [23] | - | 100 | 4.26 ms | 100 | 69.6 ms | 100 | 1.52 s |
| | | Scipy [30] | - | 100 | 5.10 ms | 100 | 148 ms | 100 | 8.75 s |
| | | lapjv | - | 100 | 5.73ms | 100 | 97.5 ms | 100 | 6.12 s |
| No Match | $\mathcal{O}(1)$ | *np.diag()* | - | 68.9 | 1.40 $\mu s$ | 67.0 | 2.49 $\mu s$ | 57.9 | 4.36 $\mu s$ |
| Approximation | $(\sqrt{s}/b) \cdot \mathcal{O}(b^3)$ | Batch-Optimal | 128 | 97.6 | 2.41 ms | 94.5 | 9.08 ms | 92.8 | 43.4 ms |
| | | | 256 | 100 | 4.56 ms | 96.8 | 16.9 ms | 95.0 | 95.9 ms |
| | | | 512 | 100 | 4.57 ms | 98.6 | 37.1 ms | 96.8 | 171 ms |
| | | | 1024 | 100 | 4.57 ms | 100 | 79.5 ms | 98.2 | 391 ms |

exists to quantify if a representational similarity metric is *better* than another. Previously Kornblith et al. [12] evaluated CKA by showing it was better at finding layers of the same architecture than SVCCA and PWCCA. This was extended by Ding et al. [3] through the inclusion of statistical testing but remains a rather shallow benchmark. Subsequently, we constrain ourselves to qualitative experiments, leaving quantitative testing to potential future benchmarks.

In all following experiments, all representations are extracted globally and zero-centered along the sample dimension. Given the zero-centered representations spatio-semantic and semantic RSMs are computed in mini-batches of 250 samples. To calculate the semantic RSMs on CIFAR an optimal bipartite matching algorithm is used, while for Tiny-ImageNet and ImageNet we utilize the *Batch-Optimal* approximation with window size $b$ 512.

### G.1 CKA between semantic and spatio-semantic RSMs

As initial inspection, we evaluate how different the similarity structures of a model measured through spatio-semantic RSMs are to a model measured through semantic RSMs. We do this by constructing both RSMs **from the same representations** of a model. Subsequently, we compare the two alternative RSMs of the same representations to each other through CKA Eq. (2). The diagonal of this matrix represents a direct comparison of identical representations, just with another definition of what is considered "similar". This is evaluated for three ResNet18s and three ResNet34s on Tiny-ImageNet with the linear and RBF kernels respectively.

We display the average CKA matrices across architecture seeds for both architectures, as well as the diagonal values of the CKA matrix. Results are shown in Fig. 15.

Examining these CKA matrices multiple observations can be made:
A) Despite the diagonal representing a comparison between identical representations, the CKA values are not 1. This indicates that the different way of constructing RSMs changes the perceived similarity structure of the system, as measured by CKA.
B) Inspecting the diagonal shows, that earlier layers with greater spatial extent express higher differences in similarity, whereas layers at a later layer and lower spatial extent are less dissimilar. This is consistent with the expectation that, with shrinking spatial extent, alignment of semantic concepts gets more likely.

Given these large changes in CKA similarity, we conclude that the definition of what a model perceives as *similar* can highly influence inter-system similarity. This is especially relevant when comparing systems across domains, where RSM construction may be domain-specific, disallowing to be consistent with RSM construction. Exemplary when comparing ML vision systems to human vision models, in particular when comparing representations of high spatial extent.

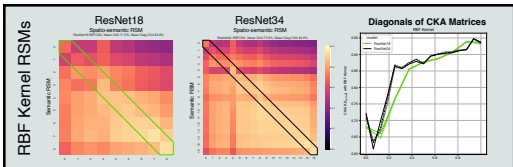 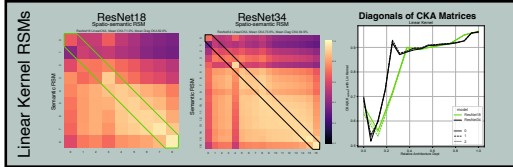

Figure 15: **The dissimilarity between *semantic* and *spatio-semantic* RSMs decreases with shrinking spatial extent.** Comparison of *semantic* and *spatio-semantic* Representational Similarity Matrices (RSMs) for the same model. The dissimilarity in early layers decreases with decreasing spatial extent, as illustrated by Centered Kernel Alignment (CKA) values. The left and middle panel shows the CKA comparison between all layers, while the right panels visualize the heatmap's diagonal, emphasizing the evolving similarity trend from early to late layers.

## G.2 Differences in CKA self-similarity

In the previous paragraph, semantic RSMs were directly compared to spatio-semantic RSMs. While this can influence measurements when models are compared across domains (e.g. CNN Vision to Biological Vision), it does not need to imply that the CKA similarity of vision models changes substantially.

Given that within the same domain, the RSM construction will likely be chosen consistently, one can opt to either: Calculate only spatio-semantic RSMs or only semantic RSMs, due to personal opinions or preferences. Subsequently, the question is not *"Are spatio-semantic RSMs similar to semantic RSMs"*, but *"Does the CKA similarity between spatio-semantic RSMs change when calculating semantic RSMs"*.

To address this question we: A) Compare the CKA matrix difference of the CKA matrix based on semantic RSMs to the CKA matrix of spatio-semantic RSMs when comparing a model with itself (Intra Model) and B) when comparing between different models (Cross-Model).

**Intra-Model CKA** Similarly to before, we extract representations and form semantic and spatio-semantic RSMs. We extract representations on CIFAR100 [14] ($32 \times 32$), Tiny-ImageNet ($64 \times 64$) and ImageNet-1k [2] ($160 \times 160$) datasets, from 3 differently seeded and trained from scratch ResNet18, ResNet34 and ResNet101 architectures. The semantic RSMs are calculated utilizing the $Batch - Optimal$ matching with $b$ 512 for matching on Tiny-ImageNet and ImageNet. We calculate semantic and spatio-semantic RSMs with a mini-batch size of 250, subsequently using them for Canonical Correlation Analysis (CKA) calculations. The corresponding cka matrices and their differences are displayed in Fig. 16. If not further specified the experiment uses the linear kernel.

Introspecting the results it can be seen that across all Architectures ResNet18, ResNet34, and ResNet101 largely the same change in similarity structure can be observed. For CIFAR100, the very first layers are perceived as less similar to semantic RSMs than with spatio-semantic RSMs, while the CKA between the middle to later layers is more similar. This structure, though does not remain consistent across datasets: When moving from CIFAR100 to Tiny-ImageNet earlier layers appear to become more similar while intermediate layers become less so. On ImageNet1k CKA on semantic RSMs seem to indicate models are more similar. This trend indicates that the influence of semantic RSMs on spatio-semantic RSMs seems to be largely dataset-dependent. Moreover, the overall maximum change in CKA similarity in these matrices is between $-0.2$ and $+0.2$ for Tiny-ImageNet, indicating a modest change in overall CKA.

**Cross-Model CKA** Aside from evaluating only CKA similarity of RSMs of the same model we extend to comparing RSMs between models, as commonly done when comparing models through CKA. ResNet18/101 models trained on CIFAR100 are used with RSMs constructed identically to previously specified. Results are displayed in Fig. 17.

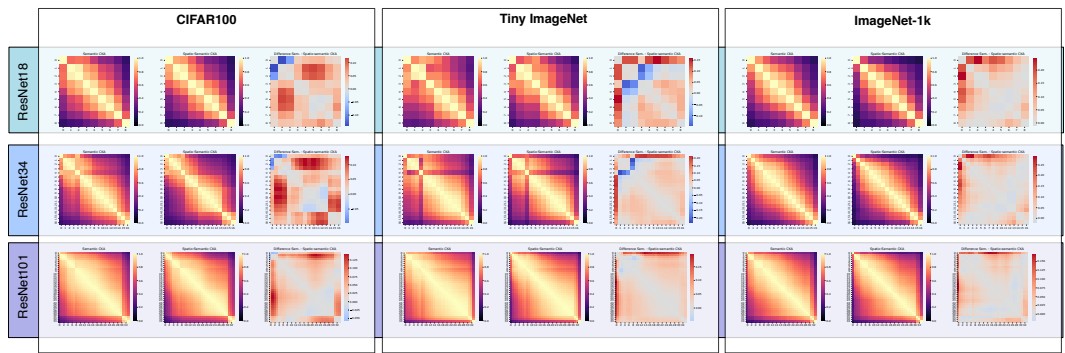

Figure 16: CKA similarity between different Layers of *the same* ResNet18, ResNet34, ResNet101, on CIFAR100, Tiny-ImageNet and ImageNet-1k. For each row the left-most CKA matrix displays the spatio-semantic RSMs, the middle represents the semantic RSMs while the right represents the difference in CKA similarity between the two. Blue regions indicate where the We observe that similarity within the block structure is largely unchanged, whereas the similarity across the later blocks seems to be more similar and the similarity of the very first blocks is less similar.

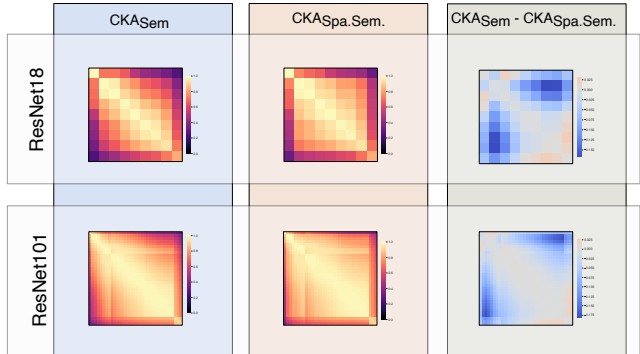

Figure 17: CKA similarity between different Layers of *different* ResNet18 and ResNet101 models on CIFAR100. We observe a decrease in similarity at high-resolution layers, whereas similarity between deeper layers is largely unchanged.

It can be seen that for both, ResNet18 and ResNet101, the cross-model CKA similarity is mostly negative for the majority of the layers, indicating that CKA on spatio-semantic RSMs estimates models to be more similar than when applying CKA on semantic RSMs. Similarly to before CKA changes range from $-0.175$ to $+0.025$ providing modest changes.

Concluding the Intra and Cross-Model CKA experiments it can be seen that the choice of RSMs results in qualitatively different CKA matrices. Unfortunately, due to the lack of quantitative benchmarks, no direct recommendation of which RSM to use for inter-model similarity calculation through CKA can be given.

