# OpenReview forum: "Decoupling Semantic Similarity from Spatial Alignment for Neural Networks."
_NeurIPS.cc/2024/Conference — NeurIPS 2024 poster_

### Official Review · Reviewer_zPy6 · 2024-07-09

**Soundness:** 3
**Presentation:** 3
**Contribution:** 3
**Rating:** 6
**Confidence:** 3

**Summary:**

In this paper, the authors propose a new method to measure similarity between responses of deep neural networks in vision. They reformulate the commonly used strategy to compute Representational Similarity Matrices (RSMs) by acknowledging the superiority of the semantic information over the spatio-semantic information in the creation of RSMs. The authors perform different experiments to show the improvement over the baseline method caused by their reformulation.

**Strengths:**

- The paper is clearly written and tackles an important topic.
- The idea is original and besides its limitation connected to high computational needs, it could serve as an inspiration for future works.
- Although the experiments are not exhaustive, they are convincing and coherent and the gained insights seem relevant.
- Being transparent with the limitation of the method and trying to provide the means to mitigate it is a plus.

**Weaknesses:**

- The authors should better discuss the differences between the results obtained for ViTs and CNNs in their study, which are quite well visible. E.g. while in the case of an examined CNN, the spatio-semantic RSM does not reflect well the similarities between translated images, in the case of the examined ViT (appendix), these similarities can be observed. The other thing is that the experiment is slightly different, because in the experiment with CNNs much smaller images are used than in the ViT experiment. Also, the differences are also visible in Table 1 (ResNets obtain much higher absolute correlation values for the baseline and the proposed methods than ViTs).
- The authors provided few visual examples of the results of their method. It would be good to provide more of them (e.g. for different similarity metrics used, for more images and for more netorks) to enable more comprehensive qualitative evaluation (they could be placed in the appendix).
- The use of some methods at work is not well justified (e.g. Pearson correlation).

**Questions:**

- Why do the authors use Pearson correlation to examine the relationship between the Jensen-Shannon Divergence and the representational similarity? E.g. Kendall/Spearman correlation can be more robust.
- The authors should focus on the differences between the results obtained for CNNs and ViTs (see the comment in the weaknesses section).
- The statement in the introduction “we argue that the spatial location of semantic objects does neither influence human perception nor deep learning classifiers.” is a little bit too bold - the paper does not examine human perception, therefore it would be better to leave only the deep learning here.
- Also, a minor thing is that some typos, grammatical and formatting errors can be found in the paper (e.g. the sentence starting in l79, l205: a RSM -> an RSM, l25: SAMor CLIPSeg ,the retrieval performance)
- The authors could provide more examples of their method for different networks to enable their better qualitative assessment which is now limited (e.g. in the appendix).

**Limitations:**

The authors adequately discussed the biggest limitation of their work (the computational cost of their method). They could also better highlight the limited data used in the experiments caused by the mentioned computational constraints.

---

> ### Author Rebuttal · Authors · 2024-08-06
>
> > _Why do the authors use Pearson correlation to examine the relationship between the Jensen-Shannon Divergence and the representational similarity? E.g. Kendall/Spearman correlation can be more robust._ and _"The use of some methods at work is not well justified (e.g. Pearson correlation)."_
>
> We have added a brief explanation in Section 4.3 regarding our choice of Pearson correlation: "We chose to use the pearson correlation, as it allows observing a direct linear behavior between representational similarity and predictive similarity." IWe believe this is particularly appropriate for cosine similarity and the radial basis function, which are bounded. However, we also provide Spearman correlation results in the appendix, along with a discussion. The corresponding table can also be found in the rebuttal PDF.
> Overall, we observe that Spearman correlation is mostly lower, with large ResNets being impacted strongly while ResNet-18 is notably consistent and stable in its correlations. ViTs interestingly show a much higher correlation for cosine and radial basis function similarities compared to Pearson correlations.
>
> >The authors should focus on the differences between the results obtained for CNNs and ViTs (see the comment in the weaknesses section).
>
> We extended our discussion on CNNs vs ViT's in the appendix, going in depth into the differences of the two translation experiments provided.
>
> > The statement in the introduction “we argue that the spatial location of semantic objects does neither influence human perception nor deep learning classifiers.” is a little bit too bold - the paper does not examine human perception, therefore it would be better to leave only the deep learning here.
>
> We cut the statement short to “we argue that the spatial location of semantic objects does not influence deep learning classifiers.”
>
> > The authors could provide more examples of their method for different networks to enable their better qualitative assessment which is now limited (e.g. in the appendix).
>
> As displayed in the rebuttal pdf, we will add plenty qualitative visualizations with a discussion in the appendix for many models. Additionally, as we added the Cityscapes experiment, we will also provide qualitative results on this dataset.
>
> >Also, a minor thing is that some typos, grammatical and formatting errors can be found in the paper (e.g. the sentence starting in l79, l205: a RSM -> an RSM, l25: SAMor CLIPSeg ,the retrieval performance)
>
> We want to apologize for this oversight, we have re-proofread our manuscript and pushed it through a spell-checker to fix any potential typos and grammatical errors.
>
> > They could also better highlight the limited data used in the experiments caused by the mentioned computational constraints.
>
> Instead of highlighting this, we increased our overall sample size, as mentioned in the main response. We hope this to be a sufficient adaptation.
>
> Find the similarity table with spearman's rank correlation below (N=20.000).
>
> | Metric          | pearson_corr |        |               |        |         |        | spearman_rank |        |               |        |         |        |
> |-----------------|-------------:|-------:|--------------:|-------:|--------:|-------:|--------------:|-------:|--------------:|-------:|--------:|-------:|
> | Kernel          |   cosine_sim |        | inner_product |        | rbf |        |    cosine_sim |        | inner_product |        | rbf |        |
> | Invariance      |            - |     PI |             - |     PI |       - |     PI |             - |     PI |             - |     PI |       - |     PI |
> | Architecture    |              |        |               |        |         |        |               |        |               |        |         |        |
> | ResNet18        |       -0.279 | -0.328 |        -0.264 | -0.272 |  -0.174 | -0.197 |        -0.231 | -0.337 |        -0.239 | -0.225 |  -0.435 | -0.476 |
> | ResNet50        |       -0.256 | -0.305 |        -0.249 | -0.269 |   0.028 |  0.015 |        -0.032 |  0.007 |        -0.046 | -0.040 |   0.128 |  0.132 |
> | ResNet101       |       -0.235 | -0.330 |        -0.211 | -0.274 |   0.076 |  0.067 |        -0.007 | -0.053 |        -0.007 | -0.077 |   0.071 |  0.068 |
> | ConvNextV2-Base |       -0.162 | -0.126 |        -0.160 | -0.184 |   0.077 |  0.050 |        -0.017 |  0.026 |        -0.013 | -0.045 |   0.143 |  0.128 |
> | ViT-B/16        |       -0.058 | -0.098 |        -0.056 | -0.031 |  -0.079 | -0.120 |        -0.013 | -0.230 |        -0.021 |  0.029 |  -0.220 | -0.313 |
> | ViT-L/32        |       -0.142 | -0.189 |        -0.143 | -0.152 |  -0.131 | -0.164 |        -0.034 | -0.276 |        -0.029 | -0.014 |  -0.335 | -0.392 |
> | DinoV2-Giant    |       -0.016 | -0.046 |        -0.016 | -0.030 |  -0.013 | -0.052 |        -0.014 | -0.037 |        -0.015 | -0.022 |  -0.015 | -0.042 |

---

> > ### Comment · Reviewer_zPy6 · 2024-08-12
> >
> > I would like to thank the authors for their response along with some additional results and wish them good luck!

---

### Official Review · Reviewer_xk1Y · 2024-07-12

**Soundness:** 3
**Presentation:** 3
**Contribution:** 3
**Rating:** 6
**Confidence:** 3

**Summary:**

This paper proposes Semantic RSMs to understand the internal representations in deep neural networks. The authors argue that the current RSMs are limited by their coupling of semantic and spatial information, which restricts the assessment of similarity. The proposed semantic RSMs are spatial permutation invariant and focus solely on semantic similarity. The proposed method is shown to enhance retrieval performance and provide a more accurate reflection of the predictive behavior of classifiers.

**Strengths:**

1. This paper is well-written and easy to follow.
2. The introduction of semantic RSMs is a significant contribution, potentially leading to more meaningful comparisons between neural network models.
3. The empirical demonstration of improved retrieval performance using semantic RSMs is convincing and adds practical value to the theoretical development.

**Weaknesses:**

1. While the paper does highlight the high computational complexity as a limitation, it would benefit from a more detailed discussion on the scalability of the proposed method to larger models and datasets and the approximation error.

**Questions:**

I'm not an expert in this field, so I tend to start by looking at what other reviewers think of the paper.

**Limitations:**

Yes, the authors have addressed the limitations.

---

> ### Author Rebuttal · Authors · 2024-08-02
>
> We appreciate the time and effort you invested in reviewing our manuscript.
>
> Regarding your comment that _"it would benefit from a more detailed discussion on the scalability of the proposed method to larger models and datasets and the approximation error."_
>
> We addressed a part of this regarding scaling to larger datasets in the general response already. Regarding the question of scaling to larger models, the scaling of our method is just dependent on representation size, while model size plays no role at all.
> We add section to discuss the approximation error at the end of the Section 4.4: We cite: _"The fastest of the Batch-Optimal approximation methods shows $<8\%$% error while improving run-time $\times 36$ relative to the fastest optimal algorithm for spatial extent $4096$, while no spatial alignment shows $42\%$% deviation ["or approximation error"] from the optimal matching."_
>
> We hope that this helps to improve clarity on the approximation error quality of the approximations.

---

### Official Review · Reviewer_sf44 · 2024-07-14

**Soundness:** 2
**Presentation:** 2
**Contribution:** 2
**Rating:** 5
**Confidence:** 3

**Summary:**

The authors introduce semantic RSMs, which are designed to be invariant to the spatial arrangement of elements within images. These semantic RSMs assess similarity by treating the problem as one of set-matching, where the focus is on matching semantic content rather than spatial details. This approach not only aligns more closely with human perception but also improves the relevance and accuracy of image retrieval tasks. The paper claims that semantic RSMs offer a more robust measure of similarity by comparing them to traditional spatio-semantic methods.

**Strengths:**

- The focus on semantic content rather than spatial arrangement aligns more closely with human perception, potentially leading to more intuitive and relevant comparisons of neural network responses.

-  By being invariant to spatial permutations, this method can effectively compare images where the same objects appear in different locations.

- Semantic RSMs can be used as drop-in replacements for traditional RSMs.

**Weaknesses:**

- Employing algorithms like Hungarian matching to find the optimal permutation matrix can be computationally expensive.

- The effectiveness of this approach relies heavily on accurate identification and parsing of semantic concepts within images, which can be challenging in complex scenes or under conditions of visual ambiguity.

- While focusing on semantic content is generally advantageous, completely ignoring spatial information can sometimes omit useful contextual cues that contribute to overall image understanding. For example,

[contextual cues] a picture of a dining table with plates, utensils, and food arranged in a specific way might convey a meal setting, which could be lost if the spatial relationships are ignored.

[object interactions] Images where interactions between objects are important, such as a cat sitting on a mat, might lose their interpretative meaning if spatial information is disregarded. The semantic content (cat, mat) remains the same, but the relationship changes based on their arrangement.

[abstract content] In abstract art or images with non-literal interpretations, spatial composition itself can carry meaning and affect how the content is perceived and classified.

**Questions:**

- How well does the method scale to very large datasets or to more complex neural networks that handle highly varied or abstract visual content?

- How does the method perform under noisy conditions or when semantic parsing is imperfect due to occlusions or poor image quality?

**Limitations:**

- To better evaluate the impact of semantic RSMs, a set of diverse metrics should be established: Evaluate semantic RSMs against traditional spatio-semantic RSMs and other state-of-the-art similarity measures to highlight the improvements or shortcomings.

- The authors should explicitly state the primary objectives of employing semantic RSMs. Show that semantic RSMs can make neural network decisions more interpretable by aligning more closely with how humans perceive images.

- Apply semantic RSMs in specific use cases like medical imaging, satellite image analysis, and autonomous driving where ignoring spatial arrangements can be particularly detrimental or beneficial, providing a nuanced view of their applicability.

---

> ### Author Rebuttal · Authors · 2024-08-02
>
> Thank you for the detailed feedback. We can see that a great deal of time and thought went into it. However, we believe there may be a few misunderstandings that we would like to clarify:
>
> ### Questions
> **Q1: How well does the method scale to very large datasets[...]**
>
> R1: The scaling behavior depends on the _spatial extent_ of representations and the _use-case_ in which the method is applied. Details on this can be found in the common response.
>
> **Q2: How well does the method scale [...] to more complex neural networks that handle highly varied or abstract visual content?**
>
> R2: We are not quite sure what you mean by _"more complex neural networks of varied/abstract visual content"_. Our experiments already apply the method to various state-of-the-art networks (CLIP, DINOv2, SAM, ResNets, ConvNeXT), which are capable of handling diverse visual content. Additionally, we evaluate our method on standard ImageNet data as well as on EgoObjects, which we believe qualifies as a more complex dataset aimed at robotics.
>
> **Q3: How does the method perform under noisy conditions or when semantic parsing is imperfect due to occlusions or poor image quality?**
>
> R3: This is an interesting question! Poor conditions or visual ambiguity can degrade representations if the network cannot handle them, potentially harming, for example, the retrieval performance of our method. However, **this is not a unique issue of our method but of all methods using learned representations, such as spatio-semantic RSMs**. Even standard cosine similarity on global representations would likely experience similar degradation.
>
> ### Limitations
>
> 1. _"Apply semantic RSMs in specific use cases like medical imaging, satellite image analysis, and autonomous driving where ignoring spatial arrangements can be particularly detrimental or beneficial, providing a nuanced view of their applicability."_
>
> We added an additional experiment for autonomous driving using the Cityscapes dataset, demonstrating improvements of semantic RSMs over spatio-semantic RSMs in this context.
>
> 2. To better evaluate the impact of semantic RSMs, a set of diverse metrics should be established: Evaluate semantic RSMs against traditional spatio-semantic RSMs and other state-of-the-art similarity measures to highlight the improvements or shortcomings.
>
> In this paper, we compared the current, most commonly used methods for constructing RSMs, as introduced in the [CKA paper](https://arxiv.org/pdf/1905.00414) by Kornblith et al. (Inner Product and RBF spatio-semantic RSMs), with the addition of cosine similarity. We also introduced two metrics to assess the “goodness” of RSMs, which we believe are sufficient to demonstrate the benefits of our proposed method.
>
> ### Weaknessses
> 1. **[Runtime of Hungarian matching]**: We clearly and transparently state the runtime as a limitation in our paper and provide some algorithms to alleviate this. Reviewer zPy6 even commends us on this. To put our compute time into perspective: In representation similarity analysis many much more slow to compute metrics exist, e.g. the [IMDScore](https://openreview.net/forum?id=HyebplHYwB) which takes >12 hours for a single layer-to-layer comparison between two representations of [10.000x2048] samples using 32 CPU cores we ran in a different project or the RSMNormDifference which can take multiple hours as well. The former was published in ICLR20. This is somewhat of a duplication to the common response,  but we felt it necessary to reiterate.
> 2. **[Visual ambiguity can negatively impact]**: See Quesions R3
> 3. **[Ignoring spatial context can be bad]:** We really appreciated these thought experiments as they go into a lot of detail on how representations work.
>    1. **[Contextual cues]**: Networks progressively combine/merge information to build semantic concepts. E.g. Ears, eyes and whiskers in a neighbourhood, could lead to expression of a cat's head and so on. Similarly we expect a network to learn associations between dishes and cutlery on a table to a meal setting. If the network learns this association, breaking the spatial location with our method is not an issue, as the semantic concept vector will not only carry `<plate>` but `<plate, part_of_meal_table>` information somewhere.
>     2. **[abstract content]**: Distance and composition can carry meaning, but if the network learns to represent them similarly, this can be beneficial. For example, if you take a picture of an abstract painting, move back 5 meters, and take another picture, the spatial composition shifts despite the image being identical. Similarly, moving sideways changes the angle. If the network still represents the same semantic meaning despite these shifts, it can still be recognized.
>     3. **[object interactions]** **(Very related to context cues)** We believe that networks merge semantic information hierarchically within a neighborhood, forming increasingly abstract semantic concepts. Due to this spatial proximity, representations of a cat on a mat are encoded similarly, leading to more frequent retrievals of cats on mats, even when spatial constraints are ignored. Evidence for this is provided by CLIP models, which use global aggregation of their outputs—disregarding spatial relations—to generate final embeddings. This method effectively identifies related images and captions, such as “a cat sitting on a mat,” and we assume this approach is applicable to our setting as well.
>
> We hope that our responses have clarified the open questions and if we answered the questions sufficiently, it would be great if you could consider increasing your rating.

---

> > ### Comment · Reviewer_sf44 · 2024-08-13
> > **Post-rebuttal response**
> >
> > Thanks for the clarification.
> >
> > The paper focuses on computing representational similarity by comparing semantics with location similarity. Instead of considering both simultaneously, the proposal focuses solely on semantics. This is achieved by extracting semantic vectors from the feature tensor and comparing them as a set, rather than matching the vectors individually in spatial terms.
> >
> > The concept is straightforward, but the application is unclear, and the derived insights may not be sufficient for publication. Here are the remaining concerns:
> >
> > - What are the derived insights from the proposal? e.g., Do ViT and CNN exhibit different behaviors? What can we learn from the results? A deeper analysis is needed beyond just showing that the proposal can predict output probabilities (e.g., Table 1).
> >
> > - There are counterexamples where neglecting location information leads to issues (e.g., visual anagrams), where semantically different images might not be distinguishable by the proposal since it matches features at a set level.
> >
> > - Also, how can this proposal be applied in complex scenarios, such as when the data contains multiple objects of the same class?
> >
> > - Last but not least, the permutation matrix required for set-matching introduces high computational overhead, which could limit the applicability of the proposal to high-resolution data.

---

> ### Author Response · Authors · 2024-08-14
> **Discussion response**
>
> Thank you for engaging with our rebuttal. We are pleased that we have addressed most of your initial concerns. Regarding the remaining issues:
>
> >What are the derived insights from the proposal? e.g., Do ViT and CNN exhibit different behaviors? What can we learn from the results? A deeper analysis is needed beyond just showing that the proposal can predict output probabilities (e.g., Table 1).
>
> We believe our paper provides a plethora of insights and is not limited to "just show[ing] that the proposal can predict output probabilities". In our paper, we:
> 1. Highlight that current RSM construction is flawed and has limitations of requiring spatial alignment, which has not been adequately discussed previously.
> 2. Introduce an algorithm that addresses this by utilizing set-matching, offering a novel approach.
> 3. Demonstrate the utility of our method through two applications, revealing significantly improved retrieval performance with five general-purpose feature extractors across the EgoObjects and Cityscapes datasets, and better correlation between the Jensen-Shannon Divergence of output probabilities and inter-sample similarity.
>
> We believe our work underscores a critical gap in current RSM-based methods and sets the foundation for further exploration into RSM construction, effective inter-sample similarity measures, and their applications for retrieval.
>
> >There are counterexamples where neglecting location information leads to issues (e.g., visual anagrams), where semantically different images might not be distinguishable by the proposal since it matches features at a set level.
>
> We understand your concerns regarding location information. However, __i)__ our experiments on the EgoObjects dataset, which includes varied contextual scenarios such as the meal table example, indicate that our method outperforms existing RSMs. This suggests that issues related to spatial location might either be rare or of lesser importance. __ii)__ Our method is based on learned representations. _Should a visual anagram exist where similar representations are expressed by the network, we believe this is to be a failure of the model and not our approach_. We don't believe a retrieval method should try to compensate for this. __iii)__ There are other examples where spatial position is neglected entirely: E.g. the ViT paper [1] shows (in Appendix D.4) that ViT's are able to learn meaningful representations for classification without any positional embedding, making the input a bag-of-words.
>
> > Also, how can this proposal be applied in complex scenarios, such as when the data contains multiple objects of the same class?
>
> Our method has already been successfully applied in scenarios involving multiple objects of the same class. Both retrieval datasets, EgoObjects (Figure 3) and Cityscapes (Rebuttal PDF) contain multiple objects of the same class. Additionally, one currently provided qualitative example in the paper and the additional qualitative examples in our rebuttal PDF illustrate our propose methods effectiveness in exactly such complex scenarios, highlighting retrieval on images with multiple instances of glasses or bowls.
>
> >Last but not least, the permutation matrix required for set-matching introduces high computational overhead, which could limit the applicability of the proposal to high-resolution data.
>
> We acknowledge the computational demands introduced by the permutation matrix for set-matching. Nevertheless, we have applied our method to high-resolution images (1920x1080) from the EgoObjects and Cityscapes datasets, which demonstrates its feasibility on large-scale data. If your concerns are aimed towards retrieval, we want to emphasize that retrieval is generally a two stage process (see [2] Section 4), with the first being a ranking by global cosine similarity and the second being a re-ranking of the top 100-400 cases which are most similar through e.g. our method. Hence such a two-stage application can enable our method to scale easily to any number of retrieval dataset size.
> Aside from this, we transparently address this limitation in our paper and have expanded on this topic in the appendix during the rebuttal phase. Moreover we'd kindly refer you to our general response which goes into more detail on the scalability of our method and would be happy to answer any more specific concerns regarding the scalability of our method should there be any remaining.
>
> We hope this addresses your concerns and look forward to any further questions you might have.
>
> [1] Dosovitskiy, Alexey, et al. "An image is worth 16x16 words: Transformers for image recognition at scale." arXiv preprint arXiv:2010.11929 (2020).
>
> [2] Cao, Bingyi, Andre Araujo, and Jack Sim. "Unifying deep local and global features for image search." Computer Vision–ECCV 2020: 16th European Conference, Glasgow, UK, August 23–28, 2020, Proceedings, Part XX 16. Springer International Publishing, 2020.

---

### Official Review · Reviewer_Xdnj · 2024-07-14

**Soundness:** 2
**Presentation:** 2
**Contribution:** 3
**Rating:** 5
**Confidence:** 2

**Summary:**

This paper makes a contribution to the construction of RSMs in the field of vision neural networks and puts forward the concept of semantic RSMs, which is innovative and theoretical.

**Strengths:**

The proposed semantic RSMs are used for spatial alignment by means of optimal permutation, which is a relatively new and promising method.

This paper verifies the validity of semantic RSMs through experiments such as image retrieval and probabilistic similarity comparison. An in-depth analysis of the experimental results is carried out, and the advantages of semantic RSMs in specific tasks are pointed out.

**Weaknesses:**

This paper lacks the experimental verification of specific downstream tasks, such as detection and segmentation, on semantic RSMs. I need to know which scenario is more suitable for RSMs and semantic RSMs.

Lack of quantitative comparative data. It is suggested to add tables or charts to show specific performance comparison data between semantic RSMs and existing methods in different tasks (such as image retrieval, class probability similarity comparison, etc.), including accuracy, time complexity and other indicators.

The discussion of the experimental results was not thorough enough. It is recommended to add a detailed analysis of the experimental results to explain why semantic RSMs perform better on certain tasks, as well as possible reasons and limitations.

 "aligns" to "align" in line 57.

**Questions:**

It is suggested to further elaborate the potential and specific scenarios of the research in practical applications to enhance readers' understanding of its practical value.

**Limitations:**

see weekness

---

> ### Author Rebuttal · Authors · 2024-08-02
>
> We would like to express our sincere gratitude for taking the time to read our paper and provide valuable feedback and constructive criticism:
>
> 1. _“This paper lacks the experimental verification of specific downstream tasks, such as detection and segmentation, on semantic RSMs. I need to know which scenario is more suitable for RSMs and semantic RSMs.”_
>
> We believe this may be a misunderstanding. Semantic RSMs are not algorithms for detection or segmentation (which we do not claim), but rather they measure the similarity between neural network representations of images. If the concern is about applying our method to dense datasets like those used for object detection or segmentation, our EgoObjects experiment (Fig. 3) demonstrates its application to a dense dataset. Additionally, the results in Table 1 show our method applied to ImageNet.
>
> 2. _"Lack of quantitative comparative data. It is suggested to add tables or charts to show specific performance comparison data between semantic RSMs and existing methods in different tasks (such as image retrieval, class probability similarity comparison, etc.), including accuracy, time complexity and other indicators."_
>
> We have added an additional retrieval experiment on the Cityscapes dataset, comparing semantic and spatio-semantic RSMs for this use case. Regarding the comparison of semantic RSMs with other retrieval methods, such as those using learned embedding spaces, we disagree that this is necessary. The aim of our paper is to find the best way to measure the similarity between representations of two samples. Parametrized methods that alter representations in non-trivial ways would make such comparisons unfair. We emphasize that the main purpose of our paper is to address the flaws in current RSM construction, which often requires spatial alignment. We demonstrate these flaws through retrieval and class probability similarity comparisons and show that our semantic RSMs alleviate these issues.
>
> 3. “It is recommended to add a detailed analysis of the experimental results to explain why semantic RSMs perform better on certain tasks, as well as possible reasons and limitations.“
>
> We add a clarifying section at the end of the retrieval section, right before 4.3 discussing the retrieval results. We cite:
> >“These experiments display clearly that demanding spatial alignment can be a significant shortcoming when semantically similar concepts are misaligned. In Fig. 4, the network learned to represent the objects very similarly, despite a shift in perspective, but due the same objects not aligning anymore, spatio-semantic similarity fails to recognize this.
> This effect should generalize to other datasets where objects are not heavily centered. For datasets with heavy object-centric behavior, like ImageNet, this should be less pronounced.”
>
> 4. Typos
> We made sure the entire manuscript has the respective errors fixed.

---

### Author Rebuttal · Authors · 2024-08-02

Thank you to all the reviewers for their time and effort in reviewing our paper. We appreciate the feedback and tried to mitigate issues to the best of our abilities. We recognize different viewpoints, but some criticisms seem based on misunderstandings, which may have led to undeservedly lower ratings. We hope to clarify and address these issues in this rebuttal.

### Added content:
While not relevant for all reviewers, we want to highlight the content added during the rebuttal as transparently as possible:
1. We conducted an additional experiment comparing semantic RSMs and spatio-semantic RSMs for retrieval using the Cityscapes dataset for autonomous driving. Results are provided in the rebuttal PDF. Our findings demonstrate that semantic RSMs are preferable to spatio-semantic RSMs for retrieval in this setting. More details are included in the figure caption, and we will offer a more comprehensive discussion in the appendix. Qualitative examples are also included but could not be fit on the single page of the rebuttal PDF.
2. We increased the sample size of our correlation experiment in Table 1 from 2,000 to 20,000 samples. Results are provided in the rebuttal PDF. Additionally, we present further quantitative results from our EgoObjects retrieval experiment, as shown in main Figure 4. We include results for 2,500, 5,000, and 10,000 database samples in table format.
3. Finally, we include additional qualitative retrieval results for various models. In the rebuttal PDF, we provide visualizations from CLIPSeg, in a downscaled excerpt.
4. Aside from this we edited small paragraphs of text, where criticised by the Reviewers.
### Common question: How well does the method scale to large datasets?
As multiple reviewers had questions about scaling behavior, we want to use this space to discuss scaling in some more depth, and put this into perspective to currently exiting similarity methods common in the representational similarity analysis space.

The scaling behavior of our poposed method depends on mostly two factors
1. Spatial Extent and Use-Case:
    1. Spatial Extent: Smaller spatial extents result in less overhead during matching. For example, ViT models, such as ViT-L/16, have a spatial extent of 257 (with 1 class token) and can be compared essentially in the same way as traditional spatio-semantic RSMs. For CNNs, the situation varies with depth. Early layers, such as those in ResNet architectures, have a large spatial extent due to downsampling (e.g., a 224x224 image downsampled 4x results in a spatial extent of 3136), making early CNN layers significantly more expensive—by a factor of 12.25. In contrast, later layers with a smaller spatial extent show more favorable scalability for CNNs.
    2.	Use-Case: When comparing neural networks using RSMs, batched calculations are common. This approach reduces computational burden and memory constraints, making semantic RSMs scale effectively for this purpose, even with larger datasets. However, for retrieval, the scaling is more costly, as each query must be compared to the entire database. Therefore, retrieval scales linearly with the size of the database being queried.

We already provide very specific runtimes in the Appendix Table 2 for our proposed approximations and the relative optimal algorithms. Given these numbers and a specific datasets and use-case in question one could estimate expected runtimes in a single threaded fashion.
To aid clarity we will add the discussion above to this section in the appendix.
Aside from runtime alone, one needs to take into account that the _dense_ embeddings of the image need to be stored for each model and image as well, which can have a substantial storage demand when e.g. trying to do retrieval on ImageNet1k.

With the runtime being such a focal point though, **we want to emphasize that the runtime of many representational similarity measures are not optimal**. E.g. there exist measures that compute much more slowly than ours, e.g. the [IMDScore](https://openreview.net/forum?id=HyebplHYwB) which takes >12 hours for a single layer-to-layer comparison between two representations of [10.000x2048] samples using 32 CPU cores, which we used in a different project or the RSMNormDifference which can take multiple hours as well. With the IMDScore having been published in ICLR20 we hope we can convince you that runtime limitations are not a disqualifying factor for such methods.

---

### Decision · Program_Chairs · 2024-09-25

**Decision:**

Accept (poster)

**Comment:**

This paper deals with extending the concept of Representational Similarity Matrices (RSM) to semantic RSM, by measuring semantic similarity between input responses via solving a set-matching problem, in order to achieve permutation invariance.  The idea is validated on image retrieval and probabilistic similarity comparisons.

It has received 2x weak accepts, 1x borderline accept, and 1x borderline reject.  Reviewers have major concerns on lacking quantitative comparative data, lacking thorough discussions and analysis of experimental results, computational costs with the utilization of Hungarian algorithms, conceptual inadequacy of semantics disregarding spatial configuration, applicability under noisy semantic content detection and in complex scenarios with multiple objects of the same class, lacking more visual examples and questionable metrics etc.  The authors have provided extensive rebuttal with additional experimental results, which reviewers find mostly helpful and render an overall acceptance consensus.

Representation similarity is an ambiguous concept.   The reality lies between global feature similarity and set-matched local feature similarity for individual images.  Nevertheless, the AC recommends acceptance based on its different perspective, even at one extreme of permutation invariant matching.   Please revise the paper per reviewers' comments and additional materials in the rebuttal, strengthening the position of the paper with more convincing results and discussions.   Thank you.

~~~~~~~~~~~~~~~~~~~~~~~~
Here are extra comments from SAC.
(1) Please remove the period at the end of title. In addition, check the usage of punctuation in Line 31-33.
(2) Please use the official template. The current one suffers from the issue of paragraph spacing.
(3) It would be nice to adjust orphan lines.
(4) Please coordinate the font sizes in figures, where some fonts are too large in Figure 1 and some fonts are too small in Figure 2.